# WBCAtt: A White Blood Cell Dataset Annotated with Detailed Morphological Attributes

**Satoshi Tsutsui,**      **Winnie Pang,**      **Bihan Wen**
Nanyang Technological University
Singapore
{satoshi.tsutsui, winnie.pang, bihan.wen}@ntu.edu.sg

## Abstract

The examination of blood samples at a microscopic level plays a fundamental role in clinical diagnostics. For instance, an in-depth study of White Blood Cells (WBCs), a crucial component of our blood, is essential for diagnosing blood-related diseases such as leukemia and anemia. While multiple datasets containing WBC images have been proposed, they mostly focus on cell categorization, often lacking the necessary morphological details to explain such categorizations, despite the importance of explainable artificial intelligence (XAI) in medical domains. This paper seeks to address this limitation by introducing comprehensive annotations for WBC images. Through collaboration with pathologists, a thorough literature review, and manual inspection of microscopic images, we have identified 11 morphological attributes associated with the cell and its components (nucleus, cytoplasm, and granules). We then annotated ten thousand WBC images with these attributes, resulting in 113k labels (11 attributes x 10.3k images). Annotating at this level of detail and scale is unprecedented, offering unique value to AI in pathology. Moreover, we conduct experiments to predict these attributes from cell images, and also demonstrate specific applications that can benefit from our detailed annotations. Overall, our dataset paves the way for interpreting WBC recognition models, further advancing XAI in the fields of pathology and hematology.

## 1   Introduction

The microscopic examination of human blood samples is essential in clinical diagnostics, providing valuable insights into a wide range of health conditions. For example, white blood cells (WBCs) or leukocytes can serve as markers for various blood-related diseases in pathology and hematology. Accurate recognition of basic WBC types (neutrophils, eosinophils, basophils, monocytes, and lymphocytes) forms a fundamental component of pathological diagnostic methods [1], used to identify various blood-related conditions such as leukemia and anemia [2]. Automating the cell recognition process can significantly enhance diagnostic efficacy, and as such, multiple datasets have been proposed for WBC categorization [3, 4, 5, 6, 7]. However, these existing datasets only annotate the cell types without any morphological characteristics of each cell, a key explanatory factor in how hematologists recognize WBCs. This lack of information may limit the development of explainable artificial intelligence (XAI) for cell image analysis, which is particularly significant in the medical domain where interpretability is crucial – clinicians cannot reliably diagnose serious illnesses without clear explanations of a model's conclusions.

In this paper, we introduce WBCAtt (Figure 1), a novel dataset for WBCs that is densely annotated with morphological attributes (Figure 2). Each cell image, obtained from the PBC dataset [6], is annotated with 11 attributes, which were determined through a comprehensive process involving discussions with pathologists, literature review, and manual inspection of cell images. One example

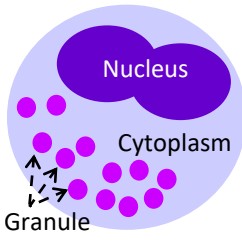
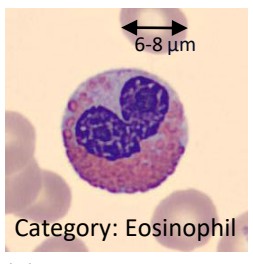

**Cell**
Size: Big
Shape: Round

**Nucleus**
Shape: Segmented bilobed
Nuclear cytoplasmic ratio: Low
Chromatin density: Densely

**Cytoplasm**
Vacuole: No
Texture: Clear
Color: Light Blue

**Granule**
Type: Round
Color: Red
Granularity: Yes

(a) Cell Structure     (b) Microscopic Image     (c) Explanatory Attributes

Figure 1: We construct a new dataset annotating 11 morphological attributes of microscopic images of white blood cells. Our set of attributes, grounded on medical literature, and plays critical role when hematologists recognize cells. Our dataset can facilitate the XAI in blood cell recognition.

of such an attribute is the presence of small cytoplasmic holes known as vacuoles, as depicted in Figure 2-(t)(u). Vacuoles provide insights into the functional state of a cell and are typically observed in monocytes and neutrophils due to their phagocytosis mechanism [8]. However, vacuoles found in other cell types, like lymphocytes and eosinophils, may indicate certain disorders [9, 10]. Based on these references, we define the presence of cytoplasmic vacuoles as one of the attributes. In the end, we establish 11 attributes, each supported by at least one medical reference. These attributes are discussed based on their association with different cellular components: overall cell (Sec. 3.1), nucleus (Sec. 3.2), cytoplasm (Sec. 3.3), and granule (Sec. 3.4). We annotated a total of 10,298 WBC images with these attributes, resulting in 113k labels (11 attributes x 10.3k images). To the best of our knowledge, this is the first public dataset to include such an extensive set of annotations, addressing the current gap in XAI for WBC analysis. Moreover, we conduct experiments to explore the capability of standard deep learning models in recognizing these attributes (Sec. 4). Furthermore, we believe that our detailed attributes can improve the interpretability of machine learning models for recognizing WBCs and related blood disorders. To illustrate this, we showcase specific applications that can be developed using our newly-introduced dataset (Sec. 5).

In summary, our contributions are as follows. We construct the first public dataset for WBCs annotated with comprehensive morphological attributes, addressing the current gap in developing interpretable models for WBC analysis. We also conduct experiments to automatically predict attributes from images, in addition to outlining specific applications. We hope that our dataset will foster advancements in XAI in pathology and hematology.

## 2 Related Work

**White Blood Cell Recognition**. WBC recognition is crucial for the diagnosis of various diseases in hematology, leading to numerous studies focusing on the development of automatic WBC classifiers [11, 12, 13, 14] and the release of publicly available datasets [3, 4, 5, 6, 7]. Prior to 2020, public WBC datasets were limited in size, containing only hundreds of images [4, 3, 5], which proved insufficient for leveraging state-of-the-art image classification models, such as Convolutional Neural Networks (CNNs). Recently, the introduction of larger datasets [6, 7], containing around ten thousand images, has enabled subsequent studies [11, 12, 13, 14] to develop novel deep learning models that improve classification performance. However, despite the inherent requirement for explainability in medical applications, no prior datasets annotate morphological attributes, which are essential explanatory factors when hematologists recognize WBCs. As such, our work aims to address this significant yet overlooked aspect.

**Attribute Datasets**. Many attribute datasets have played a pivotal role in various fields of computer vision, inspiring a diverse range of applications and fostering methodological advances. These include fashion (e.g., DeepFashion [15]) for e-commerce applications [16], human faces (e.g., CelebA [17]) for the security of facial recognition systems [18], animals (e.g., Caltech-UCSD Birds [19]) for zero-shot learning [20], and image aesthetics (e.g., AADB [21]) to investigate the inherent subjectivity of human artistic perceptions [22]. Attributes are often considered key interpretable elements in computer vision systems. For example, interpretable autonomous driving systems can be developed using explainable attributes [23, 24]. In medical domains, attributes of X-ray images [25, 26] and

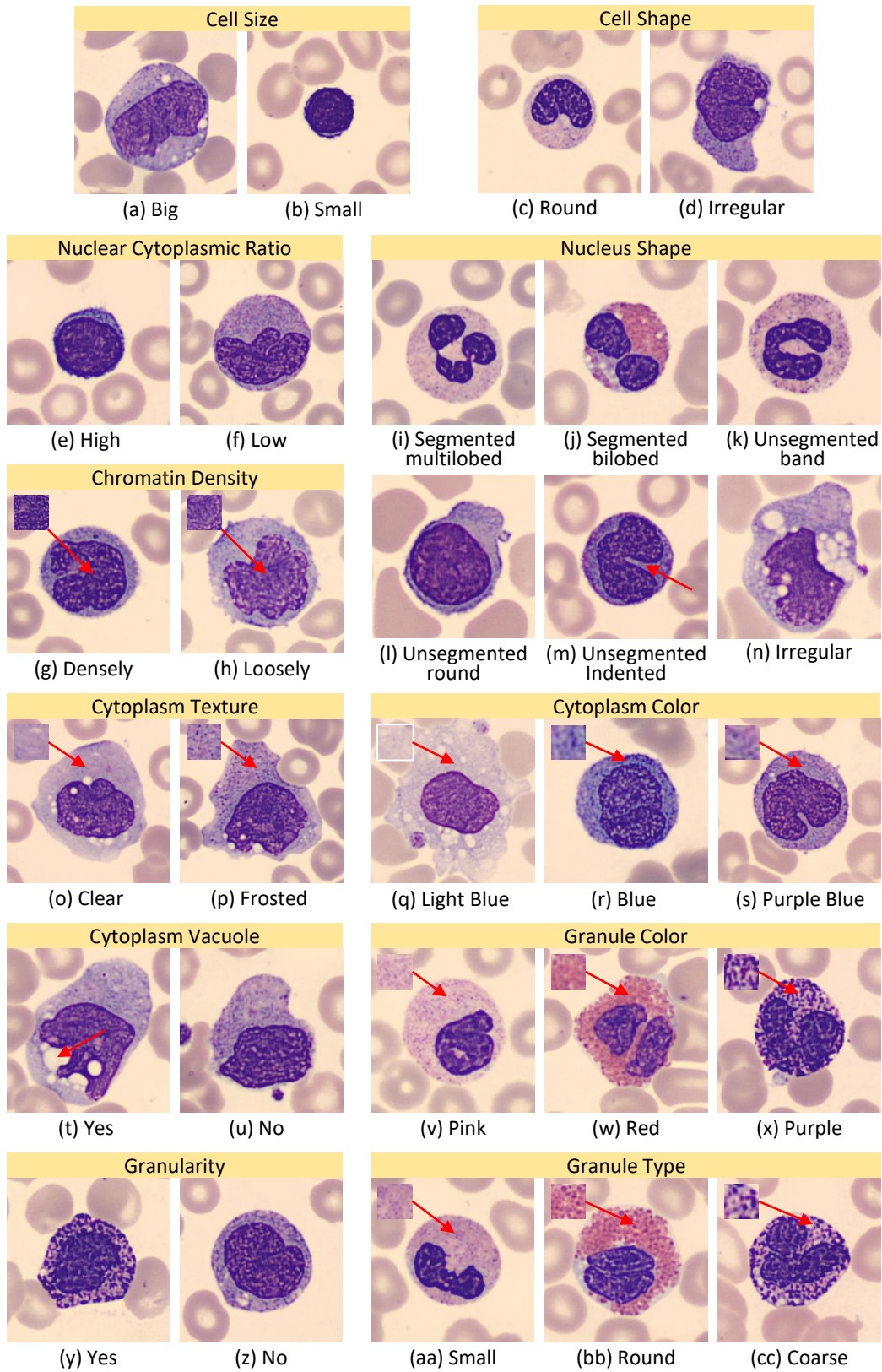

Figure 2: Sample images of each attribute. See Sec. 3 for descriptions.

skin disease images [27] have been annotated to support the development of interpretable medical AI, sharing a similar motivation with our work. However, no existing dataset provides attribute annotations for the morphological characteristics of WBCs despite their importance in hematology. Our work addresses this gap by presenting a densely-annotated dataset for WBC recognition.

# 3 WBCAtt: White Blood Cell Attribute Dataset

The attributes are categorized into four primary groups based on cell structure: overall cell, nucleus, cytoplasm, and granules, as described in Sec. 3.1-3.4, respectively. The examples of these attributes are summarized in Figure 2. We utilized all images of typical WBCs from the PBC dataset [6], which encompassed 1,218 basophils, 3,117 eosinophils, 1,420 monocytes, 3,329 neutrophils, and 1,214 lymphocytes. We annotated 11 attributes for these 10,298 images, resulting in 113,278 image-attribute pairs, with the distribution for each attribute shown in Figure 3.

**Attribute Definition Process**. Since there was no formally established set of attributes or ontology, we initiated our work with discussions with pathologists working in a laboratory at a healthcare company that develops digital cell imaging analyzers [28]. They provided five prominent attributes (see Coarse Morphological Attributes in Appendix) often used in identifying the five major WBC types, but also noted that the list was not exhaustive. Based on these initial keywords, we conducted a thorough review of relevant textbooks and research papers focusing on the morphological characteristics of WBCs. Subsequently, we refined the attribute set through further discussions with the pathologists and by manually inspecting approximately a thousand WBC images. This process yielded a total of 11 attributes, each supported by at least one medical literature reference. While the samples we inspected were the five major WBC types from healthy individuals, we expect the resulting attributes to sufficiently describe the morphological characteristics that may emerge in response to certain diseases, such as COVID-19 [29]. Moreover, these attributes provide pathologists with valuable insights into significant morphological abnormalities during microscopic examination, in accordance with the guidelines outlined in [30].

**Annotation with Quality Control**. To ensure reliable annotations across over 10k images, we devised a rigorous, iterative process involving pathologists, research scientists, and biomedical students who had strong knowledge of cell structures. In the initial stage, the students annotated the images, while being informed of the specific WBC category for each image. Following this, our research scientists meticulously examined each image and its assigned attributes, meaning that every image was inspected by at least two individuals. When ambiguities arose, we discussed with the pathologists who defined the attributes with us, ensuring a consensus on labeling. Further details on the quality control are available in the Appendix. We assessed the reliability of our annotation process by replicating it on a subset of 1,000 images with different annotators. Out of the 11,000 attribute annotations, 10,569 were consistent with the original annotations, giving an agreement rate of approximately $10569/11000 \approx 96.1\%$. This high agreement rate demonstrates the robustness and reliability of our annotation process.

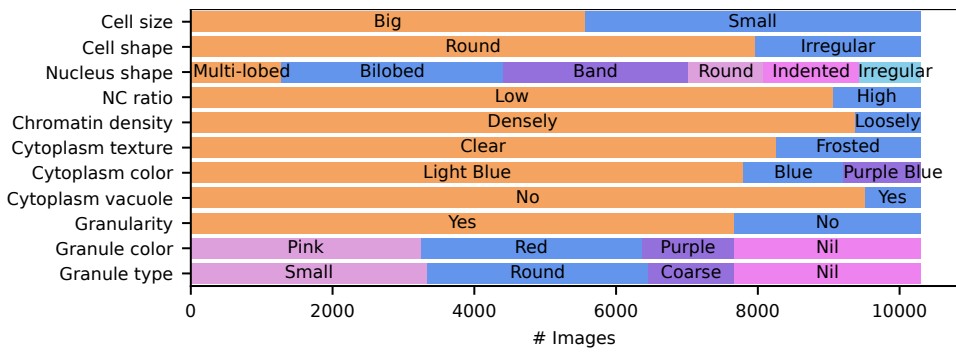

Figure 3: The distribution of values per attribute. The distribution represents the results of annotating all typical WBCs from the PBC dataset, which is the image source we utilized. We did not actively control or manipulate the distribution. See Sec. 3 for the definitions of the attributes, and Figure 2 for example images. See Appendix for the extract numbers.

## 3.1 Cell

**Cell Size**. Cell size refers to the overall dimensions of a WBC. Generally, the size of a WBC can often be indicative of its function, maturation stage, and activation state. For differentiating between various WBC types, lymphocytes usually have a small cell size compared to other WBC types, such as monocytes and granulocytes [31]. Estimating cell size is typically done by comparing the WBC to neighboring red blood cells (RBCs; usually 6-8 $\mu$m) within the same blood sample, as RBCs provide a consistent reference for size comparison. A WBC is classified as big if its diameter is larger than twice the diameter of the RBCs. Examples of big and small cells can be found in Fig. 2-(a)(b).

**Cell Shape**. The cell shape of WBCs is a significant morphological feature that offers insights into the cell type. WBCs can display a range of shapes, from round to irregular, depending on the cell type and its interactions with the surrounding microenvironment, including red blood cells. Irregular shapes are more prevalent in neutrophils or monocytes due to their unique functions or maturation stages [32, 33] Additionally, WBCs can be irregular in shape due to their interactions with adjacent red blood cells. In our definition, WBCs with circular or oval shapes are categorized as round, while any other shapes are classified as irregular. Examples of round and irregular cells are shown in Fig. 2-(c)(d).

## 3.2 Nucleus

**Nucleus Shape**. The shape of the nucleus can provide crucial information about the WBC types [34]. Segmented nuclei are typical characteristic of neutrophils and eosinophils, with neutrophils displaying a multilobed nucleus and eosinophils often exhibiting a bilobed nucleus [35]. Band-shaped nuclei can be found in immature neutrophils, also known as band neutrophils [36], which are the intermediate stage in the maturation process of segmented neutrophils. Unsegmented nuclei can be observed in lymphocytes and monocytes, with lymphocytes typically having a round nucleus and monocytes having an indented or kidney-shaped nucleus [31]. An "irregular" class is included to accommodate the nucleus shape that does not fall under the defined classes, often for the nucleus of a basophil or monocyte. In total, we have six nucleus shapes (Fig. 2-(i)-(n)): segmented-multilobed, segmented-bilobed, unsegmented-band, unsegmented-round, unsegmented-indented, and irregular.

**Chromatin Density**. The density of nuclear chromatin, which refers to the compactness of chromatin within the nucleus, is an important factor for distinguishing between different types of WBCs, such as lymphocytes and monocytes. In general, lymphocytes exhibit denser, heterochromatic nuclear chromatin, while the nucleus of monocytes appears as a "rough mesh", which contains more loosely packed, euchromatic chromatin [35, 37]. For the benefit of individuals who are not familiar with clinical terminology in using our annotations, we describe the chormatin density as loosely packed (euchromatic) and densely packed (heterochromatic), as illustrated in Fig. 2-(g)(h).

**Nuclear cytoplasmic (NC) Ratio**. The NC ratio refers to the proportion of the cell's volume occupied by the nucleus relative to the cytoplasm, and can provide valuable information regarding the cell type. A WBC with a high NC ratio is typically a lymphocyte, while a lower NC ratio is characteristic of other types. In this dataset, a WBC with an NC ratio greater than 0.7 [38] is considered to have a high NC ratio. This characteristic is often associated with a thin rim of cytoplasm surrounding the nucleus. Fig. 2-(e)(f) depicts examples of WBCs with low and high NC ratios.

## 3.3 Cytoplasm

**Cytoplasm Color**. The color of the cytoplasm can offer valuable information regarding the cell type, as different WBCs often exhibit varying cytoplasm colors due to differences in granule content and staining affinity. Furthermore, the cytoplasmic color can offer insights into granulocyte maturation, as it tends to vary across different stages of WBC development [39]. In light of this, we have annotated the cytoplasm colors ranging from light blue to purple blue [40], as shown in Fig. 2-(q)(r)(s).

**Cytoplasm Texture**. Cytoplasm texture also contributes to the classification of WBCs, as different cell types may exhibit unique textures due to variations in intracellular content, such as granules and other organelles. For example, tiny, dust-like purplish granules that sometimes appear in lymphocytes and monocytes give the cytoplasm a frosted or ground glass appearance [41] while the cells lacking these dust-like granules have a clear or transparent cytoplasm, as shown in Fig. 2-(o)(p).

**Cytoplasm Vacuole**. Discrete cytoplasmic vacuoles are often found in monocytes and sometimes in neutrophils due to their phagocytosis mechanism [8], where pathogens and cell debris are engulfed and digested. The presence of these vacuoles can help identify cell types and provide insights into the cell's functional state. Vacuoles found in other cell types, such as lymphocytes [9] and eosinophils [10], might be indicative of certain diseases and can aid in the diagnostic process. Examples of cells with and without a cytoplasmic vacuole are shown in Fig. 2-(t)(u).

### 3.4 Granule

**Granularity**. In this dataset, granularity means the presence of prominent stainable cytoplasmic granules [35] that distinguish between granulocytes (neutrophils, eosinophils, basophils) and agranulocytes (monocytes, lymphocytes). While granules are not entirely absent in agranulocytes, they are generally found in smaller quantities and are less noticeable compared to their presence in granulocytes [42]. Agranulocytes are categorized as having no granularity unless prominent granules are observed. Fig. 2-(y)(z) shows a granulocyte (granularity: yes) and an agranulocyte (granularity: no).

**Granule Color**. Granule color is a distinguishing factor among granulocytes. The granules within neutrophils, eosinophils, and basophils have different colors due to their distinct compositions, which reflect their unique immune functions. Neutrophil granules are typically pink, eosinophil granules are red, and basophil granules are purple [40, 42]. This attribute is particularly useful for distinguishing eosinophils from other granulocytes, as eosinophilic granules are uniquely stained red by eosin. This is due to the presence of cationic proteins within the eosinophilic granules, which bind to the eosin dye and give the cell its characteristic red color [43]. Fig. 2-(v)(w)(x) shows examples of them.

**Granule Type**. Granule type describes the morphological characteristics of granules in granulocytes. Neutrophils, eosinophils, and basophils each have their roles in the immune response, possessing distinct granule types filled with different substances. Neutrophils contain small, fine granules that are packed with antimicrobial proteins and enzymes, such as myeloperoxidase, lysozyme, and lactoferrin. The granules of the eosinophils are usually round and membrane-bound that compose of major basic protein. Basophils, on the other hand, possess conspicuous and coarse granules filled with histamine, heparin, and other mediators of inflammation [35]. Fig. 2-(aa)(bb)(cc) shows examples of them.

## 4    Attribute Prediction Experiments

While various applications are possible with our attributes (Sec. 5), a preliminary question emerges: How well do standard visual recognition models perform in predicting these attributes? This is a crucial question because if the model cannot even recognize these attributes, we cannot reliably develop interpretable models on top of it. To investigate this, we conduct experiments to predict the 11 attributes from WBC images.

**Data Split**. We randomly divided the dataset into 6,179 training images, 1,030 validation images, and 3,099 test images. The random division ensures that the cell-type distributions are the same in each set. The exact split is available in Appendix.

**Model**. Our baseline model predicts attribute values from image representations extracted by an image encoder of convolutional neural networks. We employ an ImageNet-pretrained ResNet50 as our choice of image encoder. The attribute prediction model comprises multi-task heads of per-attribute linear layers, with each layer corresponding to a specific attribute. We intentionally keep the model simple and provide a foundation for future work, such as designing more complex model architectures or integrating domain knowledge of WBCs (e.g., cell structure). The further details about the model (including other backbones) and its training is in Appendix and the source code.

**Evaluation Metrics**. Due to the imbalance of attribute values, we use macro F-measure, which calculates the harmonic mean of precision and recall, instead of plain accuracy for the evaluations. We run the same code three times with different seeds and report 95% confidence intervals.

**Results**. We present the results in Table 1. The baseline model achieves an average macro F-measure of $91.20 \pm 0.06\%$. Some attributes, such as granularity, granule type, and granule color, exhibit particularly high F-measures of over 98%. On the other hand, nucleus shape is the most challenging attribute to predict, with the lowest F-measure of $76.13 \pm 0.59\%$, which could be due

Table 1: Macro F-measure (%) for Attribute Prediction.

| Cell Size | Cell Shape | Nucleus Shape | Nuclear Cytoplasmic Ratio |
|---|---|---|---|
| $83.81 \pm 0.33$ | $90.66 \pm 0.36$ | $76.13 \pm 0.59$ | $96.35 \pm 0.06$ |
| Chromatin Density | Cytoplasm Vacuole | Cytoplasm Texture | Cytoplasm Color |
| $86.39 \pm 0.32$ | $89.57 \pm 0.47$ | $94.49 \pm 0.51$ | $87.99 \pm 0.47$ |
| Granule Type | Granule Color | Granularity | (Average) |
| $99.44 \pm 0.07$ | $98.76 \pm 0.08$ | $99.61 \pm 0.02$ | $91.20 \pm 0.06$ |

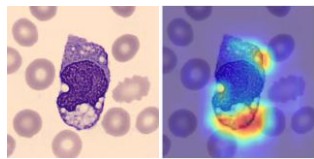 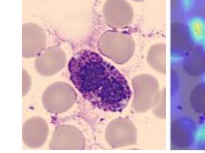 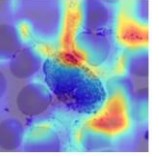 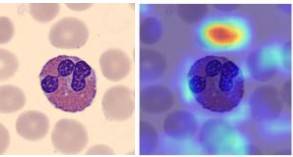

(a) Cytoplasm vacuole  GT: Yes  Pred: Yes  (b) Cell shape  GT: Round  Pred: Irregular  (c) Cell size  GT: Big  Pred: Small

Figure 4: Examples of correct and incorrect predictions with Grad-CAM. (a) Vacuoles are correctly highlighted. (b) The model incorrectly identifies leaked cellular substances as a part of the cell. (c) The model overlooks the WBC and focuses on the red blood cell instead.

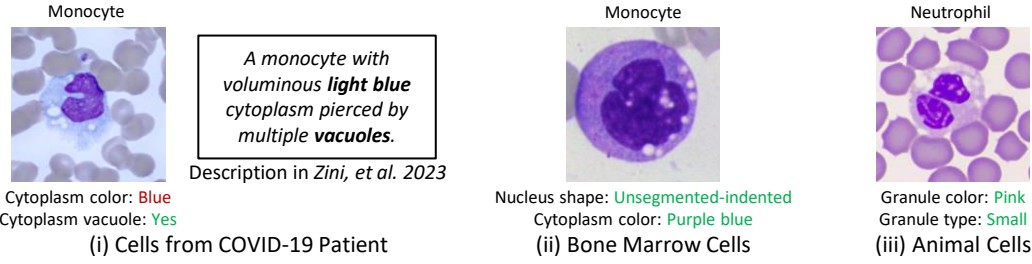

Monocyte

*A monocyte with voluminous **light blue** cytoplasm pierced by multiple **vacuoles**.*

Description in *Zini, et al. 2023*

Cytoplasm color: Blue
Cytoplasm vacuole: Yes
(i) Cells from COVID-19 Patient

Monocyte

Nucleus shape: Unsegmented-indented
Cytoplasm color: Purple blue
(ii) Bone Marrow Cells

Neutrophil

Granule color: Pink
Granule type: Small
(iii) Animal Cells

Figure 5: Examples of correct and incorrect predictions when the predictor is applied for cell images beyond our dataset: (i) Covid-19 Patients, (ii) Bone Marrow Cells, and (iii) Animal Cells. While not all attributes can be predicted correctly, especially those sensitive to staining, our attribute predictor can still recognize key attributes. See Broader Applicability in Sec. 4 for further details.

to the complexity of the nucleus shape. Figure 4 highlights some prediction results along with Grad-CAM[44] to visualize the areas the model focuses on. More examples are reported in Appendix.

**Broader Applicability**. Although we established attributes based on typical WBCs from healthy human individuals, we anticipate that our attribute definitions can be applied in diverse contexts. For example, Zini and d'Onofrio [29] discuss the morphological features of WBCs in COVID-19 patients, which align with the attributes we have established. To explore how well our classifier can recognize them, we briefly inspected a small number of images involving cells from COVID-19 patients [29], bone marrow samples [45], and swine (a non-human species) [46]. While the attributes are still applicable, we observe that the cytoplasm color, which is sensitive to staining conditions, sometimes cannot be predicted correctly. As we can see in Figure 5 in comparison to Figure 2, colors look different. Nonetheless, we find that attributes less sensitive of colors (e.g., cytoplasm vacuole) can be predicted correctly for the majority of the cases. We discuss this further in Appendix.

## 5   Applications

One immediate practical application is to automatically recognize morphological features from cell images, which is investigated in Sec. 4. The attribute recognition model can be incorporated into software that automatically analyzes cell morphology [28], which provides valuable assistance in clinical diagnostics, as certain morphological features of WBCs may indicate specific diseases or conditions [47, 48]. It can also assist hematologists in searching for cells with specific morphological characteristics within massive datasets that would be impractical to examine manually. Beyond the cell analyzer, we demonstrate three applications that can contribute to XAI.

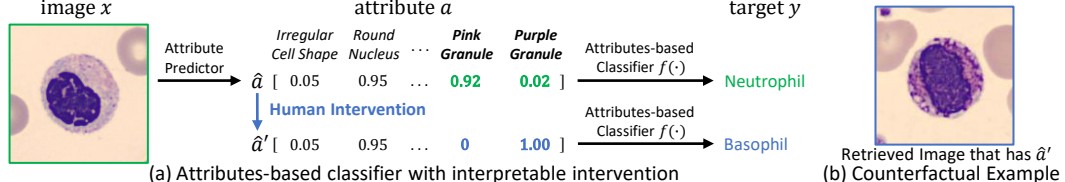

(a) Attributes-based classifier with interpretable intervention     (b) Counterfactual Example

Figure 6: (a) As discussed in Sec. 5.1, our dataset enables training a cell type classifier solely based on attributes. We can ask questions like, "What would be the predicted cell type if this cell had purple granules instead of pink?" (b) We can retrieve corresponding images as counterfactual examples.

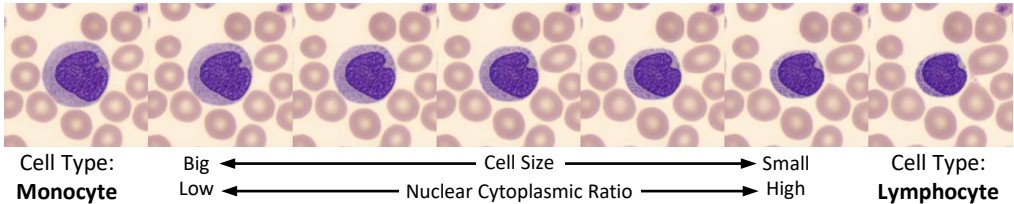

| Cell Type: | Big | Cell Size | Small | Cell Type: |
| **Monocyte** | Low | Nuclear Cytoplasmic Ratio | High | **Lymphocyte** |

Figure 7: Our dataset enables training GANs for attribute-based image editing, which can be used for synthesizing counterfactual examples. For instance, controlling cell size and NC ratio can illustrate the decision boundaries between monocytes and lymphocytes.

## 5.1 Human Intervention with Highly Interpretable Models

One of our motivations in developing this dataset is to foster XAI for WBC recognition. An effective approach to enhance explainability is designing models that make predictions based exclusively on attributes that are easily interpretable by humans. Koh et al. [49] explored such a model that initially uses a CNN to predict a set of human-interpretable attributes and subsequently uses these attributes to predict the target output. Formally, the models are trained on data points of $\{\text{image } x, \text{attribute } a, \text{target } y\}$, using $x$ to predict attributes $\hat{a}$ and then relying exclusively on $\hat{a}$ to estimate the target $\hat{y}$. We implemented a basic version of this model using the attribute predictor developed in Sec. 4. In particular, we trained a L1-regularized linear softmax classifier $f(\cdot)$ to predict WBC types from the probabilities of attributes inferred from an image. Importantly, we limit the $f(\cdot)$ to depend solely on these attribute probabilities to determine the cell category. The model can predict the WBC categories with a F-measure of $99.40 \pm 0.04\%$, whereas a CNN predicting directly from images can achieve $99.54 \pm 0.05\%$. Despite the slightly reduced accuracy, the attribute-based model facilitates more engaging human-model interactions, akin to Koh et al. [49], by enabling human interventions with the edited attributes $\hat{a}'$ and observing how this impacts the prediction $f(\hat{a}')$ versus $f(\hat{a})$. As shown in Fig. 6-(a), we can analyze hypothetical scenarios, such as what would happen if a cell contained granules of a different color. Such analysis would be infeasible without our dataset.

## 5.2 Counterfactual Example Retrieval and Synthesis

Another way of explaining a classifier is to show counterfactual examples, where a slight modification to the input image can result in a different classification outcome by the classifier. Our attribute predictor described in Sec. 4 can be utilized for retrieving counterfactual examples that correspond to attribute changes. Fig. 6-(b) displays a retrieved example where the pink granules of the cell in Fig. 6-(a) are changed to purple ones, illustrating the decision boundary between neutrophils and basophils. Furthermore, we can even generate counterfactual examples for a given WBC image by utilizing data-driven image editing techniques [50]. As a proof-of-concept, we trained an unconditional StyleGAN [51] and implemented GAN-based editing techniques [52, 53] to modify cell size and the nuclear cytoplasmic (NC) ratio, as shown in Figure 7. These attributes are crucial in distinguishing between monocytes and lymphocytes [31, 38]. Larger cells with a lower NC ratio (depicted on the left in Fig. 7) are likely monocytes, whereas smaller cells with a higher NC ratio (shown on the right in Fig. 7) are lymphocytes. These counterfactual examples helps us understand the decision boundaries.

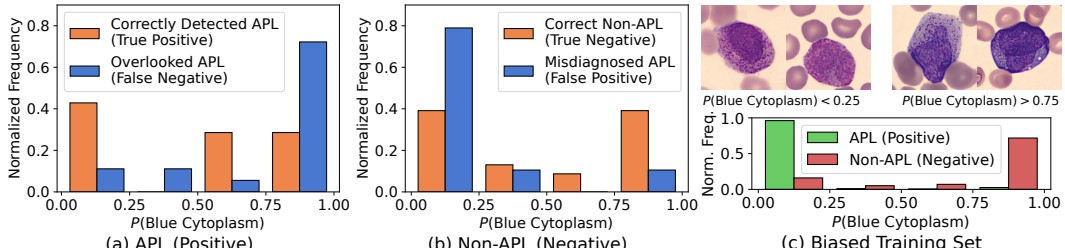

Figure 8: (*Best viewed with zoom*). We analyze the binary classifier for Acute Promyelocytic Leukemia (APL) using images of promyelocytes, a type of immature cell absent from our dataset. Employing the attribute predictor trained on our dataset, we estimate the probability distribution of morphological attributes in test images of (a): APL and (b): Non-APL. Notably, we observe a significant difference in $P$(Blue Cytoplasm) between correct predictions and incorrect predictions within (a) and (b). Specifically, plot (a) demonstrates that the classifier is likely to overlook APL if the cell exhibits blue cytoplasm, whereas plot (b) indicates the classifier tends to mistakenly diagnose APL when the cell lacks blue cytoplasm. Essentially, the classifier correlates blue cytoplasm with Non-APL. This correlation is not medically supported [60, 61] and is a coincidental correlation present in the training set (as demonstrated in plot (c)) of a specific dataset [59]. For further details, refer to Sec. 5.3.

## 5.3 Interpreting Model Bias for Blood Disorders Involving WBCs

Although our dataset annotates morphological attributes of typical cells from healthy individuals, we believe it still offers significant potential for explaining the behaviors of machine learning models for atypical cells from blood disorders. Attribute predictors trained on our dataset can unveil biases that a model might have inadvertently learned, thereby improving model interpretability. Models often misclassify images that are linked to particular attributes. This issue has been reported in various machine learning systems, including face recognition models that underperform on certain genders [54], races [55], or skin colors [56]. Similar examples have been discovered in the medical field, where skin disease detection models erroneously associate malignant cases with artificial skin markers used in clinical practices [57], or X-ray models misinterpreting the appearance of a treatment device as pathological signs [58]. By employing our attribute predictors to identify biases, we can improve model interpretability, enabling a deeper understanding of decisions and ultimately contributing to XAI that assists clinicians in adapting ML models for diagnosis.

To illustrate this point, we explore the Acute Promyelocytic Leukemia (APL) detection dataset [59] derived from peripheral blood smears. This dataset presents a binary classification task of recognizing APL from images of immature myeloid cells – an atypical category not covered in our dataset. Initially, we trained a baseline classifier using ResNet50, achieving an AUC of $76.97 \pm 2.35\%$, comparable to the reported 73.9% achieved using a 7-layer CNN. Subsequently, we applied our attribute predictors on the test images to examine the correlation between the classifier's performance and the attribute distributions in promyelocytes, the type of myeloid cells essential for diagnosing APL, as indicated by the $P$ in APL, representing *promyelocytic*, the adjective form of promyelocyte.

We explored the attribute distribution and the classifier's performance in actual APL cases and actual non-APL cases. On closer inspection of all attributes, we found that promyelocytes characterized by blue cytoplasm were frequently misclassified as false negatives, as illustrated in Fig. 8-(a). Examining the training set more thoroughly to locate the source of this bias, we found a clear correlation between blue cytoplasm and non-APL cases (Fig. 8-(c)). To validate this bias, we consulted relevant literature [60, 61] and inspected actual images within this dataset. We concluded that this was a spurious correlation present in the training set, specific to this dataset [59]. This example clearly illustrates how our dataset can help in identifying model biases, thus promoting model interpretability for XAI in medical diagnostics.

# 6  Conclusion

We have presented a densely-annotated dataset for WBC recognition, containing 11 morphological attributes for 10,298 cell images. This dataset addresses the current gap in the development of explainable and interpretable machine learning models for WBC analysis, a crucial task in hematology and pathology. We trained an automatic attribute recognition model and showcased several specific applications that can be developed using our attribute annotations. We hope that our dataset will foster advancements in XAI in the fields of pathology and hematology.

**Limitation**. We annotated images from a single source [6], which is limited to typical cells and does not include abnormal cells from blood disorders, which may be of greater clinical interest. Annotating these cells is a future direction. The cytoplasm color in our attribute definitions assume the use of the May Grünwald-Giemsa staining method, and they may appear differently with other staining methods (e.g., the color purple in granules may not look purplish). However, domain adaptation could mitigate these distributional shifts. Creating a similar dataset with other staining methods is future work.

## Acknowledgments and Disclosure of Funding

This project is supported by the Sysmex Corporation. We thank Mari Kono and Joao Nunes for helpful discussion. The research was carried out at the ROSE Lab at the Nanyang Technological University, Singapore. We thank Siyuan Yang, Takuma Yagi, and Lalithkumar Seenivasan for their valuable feedback on the manuscript. We also thank anonymous reviewers for their constructive suggestions. The computational work for this article was partially performed on resources of the National Supercomputing Centre (NSCC), Singapore (NSCC project ID: 12003626).

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

# 7 Appendix

## 7.1 Broader Impacts

Our study was exempt from institutional IRB approval as we added labels to publicly available data. We do not foresee immediate ethical concerns in our dataset as we annotate well-anonymized public data and rely on medical literature for attribute definitions. However, we acknowledge that every dataset is inevitably influenced by the potentially biased values of its creators. While the bias may be less significant in white blood cells compared to domains such as aesthetics [22], face [54], or skin [56], we remain committed to addressing any ethical concerns that may arise in the future.

## 7.2 Dataset Files

Our dataset consists of three csv files, which are hosted at `https://github.com/apple2373/wbcatt/tree/main/submission`. Moreover, the files are small enough to be permanently hosted on the NeurIPS website as supplementary material. This ensures long-term accessibility and backup. The three files of `pbc_attr_v1_train.csv`, `pbc_attr_v1_val.csv`, and `pbc_attr_v1_test.csv` contain attribute annotations for the train/val/test splits.

- `cell_size`, `cell_shape`, `nucleus_shape`, `nuclear_cytoplasmic_ratio`, `chromatin_density`, `cytoplasm_vacuole`, `cytoplasm_texture`, `cytoplasm_colour`, `granule_type`, `granule_colour`, and `granularity`: Attribute columns. We annotated them, which is the main contribution of this paper.
- `label`: One of the five WBC types (neutrophils, eosinophils, basophils, monocytes, and lymphocytes) provided by the PBC dataset.
- `img_name`: This is the image file name. It can serve as a unique identifier.
- `path`: Image path organized by the PBC dataset.

For images, please download from the PBC dataset: `https://data.mendeley.com/datasets/snkd93bnjr/1`

## 7.3 License

Our dataset is under the MIT license.

## 7.4 Details of Attribute Prediction in Section 4

### 7.4.1 Model Details

Let $M$ denote the attribute prediction model, $I$ for the input image, and $A$ for the predicted attributes. The model $M$ consists of an image encoder $E$ and an attribute predictor $P$.

The image encoder $E$ (for example, ResNet) processes the input image $I$ and generates the feature vector $F$:

$$F = E(I)$$

The attribute predictor $P$ is comprised of several attribute predictors $P_i{}_{i=1}^n$, each being a fully-connected layer tasked with predicting the $i$-th attribute. Each attribute predictor $P_i$ accepts the image features $F$ as input and outputs the predicted distribution $A_i$ for each attribute:

$$A_i = P_i(F)$$

The predicted attributes $A$ collate all the predicted attribute distributions $A_i{}_{i=1}^n$, serving as the predicted attributes for the input image $I$:

$$A = [A_1, A_2, \ldots, A_n]$$

The overall prediction of the attribute prediction model $M$ for the input image $I$ can be expressed as:

$$A = M(I) = P\left(E(I)\right),$$

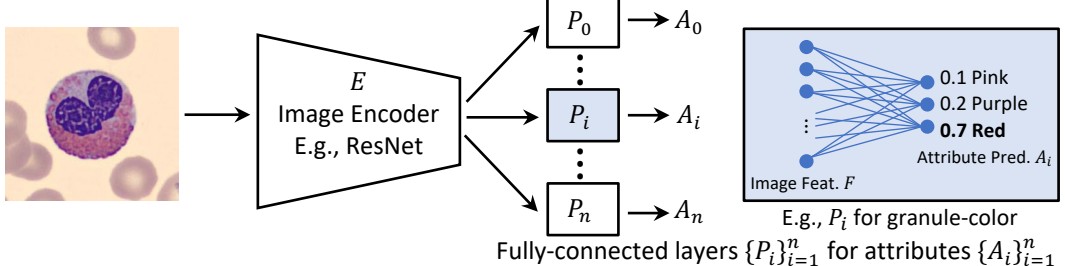

Figure 9: The architecture of the attribute prediction model. The model accepts an input image, which is processed by an image encoder $E$ to yield a feature vector. This vector is then processed by multi-task attribute predictors $\{P_i\}_{i=1}^{n}$ to produce predicted attribute distributions $\{A_i\}_{i=1}^{n}$.

where $P = [P_1, P_2, \ldots, P_n]$ represents the composition of the attribute predictors.

The architecture of this model is illustrated in Figure 9. Training this network inherently involves multi-task learning, where each task is to predict a specific attribute. The model takes an input image, which is processed by a shared image encoder to yield a feature vector. This feature vector is then utilized by several attribute predictors $P_{i\,i=1}^{n}$, each responsible for generating predicted attribute distributions $A_{i\,i=1}^{n}$.

### 7.4.2 Implementation Details

We use the Adam optimizer with a learning rate of 0.0001, a weight decay of 0.01, and a batch size of 128. We train the model for 30 epochs, selecting the best-performing model based on the validation set, and evaluate its performance using the test set. The PyTorch code for this model is available in `attribute_predictor.py`, which contains the implementation of the `AttributePredictor` class. The code to execute the experiment is `traineval.py`.

### 7.4.3 More Baselines

The main paper presents results utilizing the backbone of ResNet50 [62], which is one of the most frequently used image classifiers and has been reported as having received the highest number of citations in the field of artificial intelligence [63]. As mentioned in the main paper, we have intentionally kept the model simple. We believe that reporting results with this widely used backbone is beneficial as it provides a solid foundation for future work. However, in Table 6, we also have results from other backbones such as VGG16 [64], ViT-Base/16 [65], and ConvNeXt-Tiny [66].

### 7.5 Annotation Process and Quality Control

We used Label Studio [67] (see Figure 10) as the annotation tool. To ensure the accuracy and reliability of the annotations in our dataset, we implemented a comprehensive quality control process. This involved multiple steps and the participation of domain experts. The following sections outline the key aspects of our annotation quality control.

(i) **Recruiting Qualified Annotators**. We called for students majoring in biomedical sciences who claimed to have a basic knowledge of WBCs. We requested each of them to provide a short description of the five types of WBCs to ensure that they possessed the required foundational knowledge. We invited those who provided satisfactory answers to an information session, where we clearly explained the morphological attributes using materials similar to those presented in the main paper (Sec. 3.1 - Sec. 3.4) and Figure 2. We then recruited them to perform a pilot annotation of 100 images (see next).

(ii) **Pilot Annotation**. After the aforementioned screening process, we instructed each candidate to independently annotate a randomly selected subset of the same 100 images. We intentionally used the same images to evaluate their understanding and enable comparisons among the candidates. We acknowledge that the candidates annotated the images while being aware of the cell type. This was a necessary compromise we made by involving medical students instead of hiring pathologists due to cost limitations. The cell types in the PBC dataset had already been verified by pathologists, and this knowledge aided the students in producing higher-quality annotations. For instance, they could

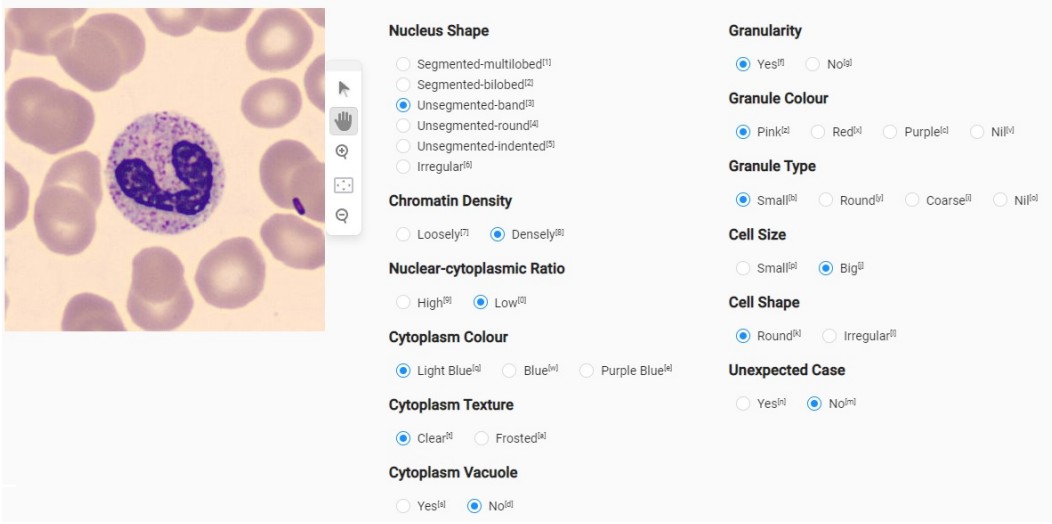

Figure 10: The screenshot of Label Studio, the labeling tool used to annotate the WBCAtt dataset. Annotators can determine the cell type from the name associated with the cell, which is provided before image selection. The image to be annotated is presented in a clear and organized manner, enabling annotators to easily navigate and zoom in on the image. Attribute selections are organized using radio buttons, ensuring that only one choice can be selected at a time. All attributes must be selected before submitting and proceeding to the next image.

utilize the cell type as prior knowledge, such as understanding that lymphocytes from healthy patients should not possess a multi-lobed nucleus.

(iii) **Annotation Phase 1 and Feedback**. Based on their performance on the pilot annotation, we selected the students and assigned them 200 images for what we refer to as Phase 1. In this phase, each student annotated a unique set of images, with allocation decisions taking into account their individual strengths. For instance, if a student demonstrated superior proficiency in annotating neutrophils compared to other cell types, we allocated a higher number of neutrophil images to that student. Subsequently, we conducted a thorough review of the annotations, providing specific feedback and updating our attribute descriptions to improve clarity.

(iv) **Annotation Phase 2 with Regular Discussions**. After completing Phase 1, we allocated the remaining images to the annotators and encouraged them to report any uncertain cases for further discussion. In cases of ambiguity, we discussed with the pathologists who defined the attributes with us, ultimately reaching a consensus. These discussions and consensus-building efforts were instrumental in ensuring the consistency and high-quality of the annotations.

(v) **Review and Validation**. Upon completion of Phase 2, our research scientists, who were not directly involved in the annotation process, meticulously reviewed and validated all annotations for correctness and adherence to the attribute definition. This means that each image in our dataset is reviewed at least by two individuals. We corrected errors or inconsistencies through this refinement process, thereby enhancing the quality and accuracy of the annotations.

(vi) **Reliability Analysis**. To assess the reliability of our annotations, we randomly selected a subset of 1,000 images, and replicated our annotation process with different annotators. Out of the 11,000 (= $11 \times 1,000$) attribute annotations, 10,569 were consistent with the original annotations, resulting in an agreement rate of approximately 96.1%. Details of the per-cell agreement rate and the per-attribute agreement rate are also tabulated in Figure 11. This high agreement demonstrates the reliability and robustness of the annotations.

### 7.6 Case Studies of Attribute Prediction

As mentioned in the main paper, we examine specific prediction results, both correct and incorrect ones, produced by our attribute prediction model. We use Grad-CAM to highlight the areas the model

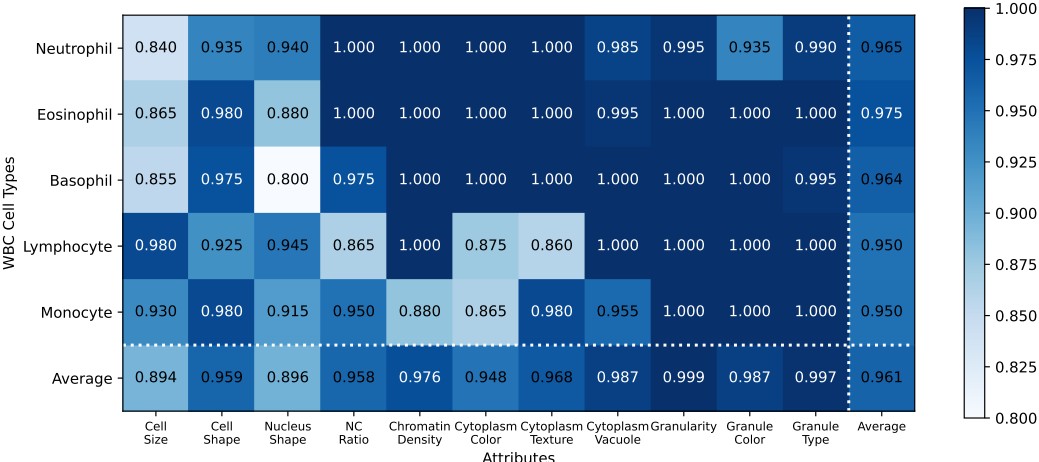

Figure 11: The inter-annotator agreement on a random sample of 1000 images from the WBCAtt dataset. Approximately 96.1% of the annotations showed consistency between the original annotations and the re-annotations by different annotators. The last column represents the per-cell agreement rate, while the last row corresponds to the per-attribute agreement rate.

focuses on. In this appendix, we present additional cases that were not included in the main paper due to space limitations.

### 7.6.1 Successful Cases on Our Dataset

When the model correctly predicted attributes, we expect that the Grad-CAM heatmap will emphasize the cell structure in line with the attribute definition. For instance, if the model is predicting the nucleus shape, it should highlight the nucleus prominently. Figure 12 shows two examples of successful attribute predictions with Grad-CAM heatmaps. In these instances, the model effectively localizes the attributes. The Grad-CAM heatmaps focus on the cell edges when predicting cell shape and cell size, as shown in Figure 12-(b), (c), (i), and (j). At times, the localization is extremely precise. Particularly, for the nucleus-shape prediction in Figure 12-(d), the Grad-CAM heatmap accurately marks the thin filament of the nucleus, which serves as a key determinant for segmented nucleus identification. Moreover, the Grad-CAM can localize the cytoplasm-vacuole, as shown in Figure 12-(m), and highlight the cytoplasmic area during the prediction of cytoplasm color, as depicted in Figure 12-(n).

### 7.6.2 Failure Cases on Our Dataset

Subsequently, we investigate the cases where the model incorrectly predicted the attributes. We expect the Grad-CAM heatmaps to indicate the reasons behind the model's failures. After manually examining multiple instances of failed predictions, we have identified the following potential factors contributing to incorrect predictions:

(i) **Presence of Two Cells in an Image**. The existence of multiple distinct cell types within an image can present a challenge for the attribute predictor. Given that attributes are highly specific to different cell types, this situation can cause confusion during attribute prediction. For example, in Figure 13-(i)(a), both a neutrophil and a lymphocyte are present in the same image. The lymphocyte is considered the ground truth for the cell type in this image, as annotated by the creator of the PBC dataset (not by us), primarily because the lymphocyte occupies a more central position. Consequently, our attribute annotations are based on the lymphocyte rather than the neutrophil. During attribute prediction, even though the Grad-CAM heatmaps focus on the lymphocyte when predicting cell-shape (Figure 13-(i)(b)) and cell-size (Figure 13-(i)(c)), the attribute predictor looks at the neutrophil when predicting other attributes such as granularity (Figure 13-(i)(e)), granule color (Figure 13-(i)(f)), and granule type (Figure 13-(i)(g)). These attributes related to granules are typically not triggered for lymphocytes in most cases, resulting in incorrect predictions.

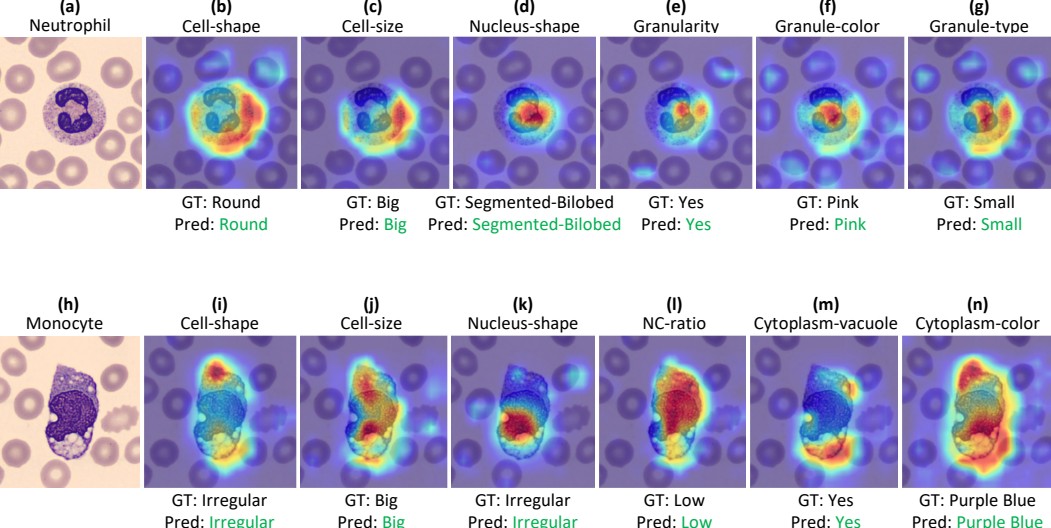

Figure 12: Grad-CAM heatmaps of accurate attribute predictions for a neutrophil ((a)-(g)) and a monocyte ((h)-(n)). Among the 11 attributes, we have selected six attributes that effectively differentiate between the respective cell types (neutrophil and monocyte).

(ii) **Broken Cell** vs (iii) **Leaked Cellular Substances**. Sometimes the structure of cell is broken. Figure 13-(ii) shows an broken cell where the cell membrane is unrecognizable, thus making the cell-shape irregular, which is correctly predicted. On the other hand, Figure 13-(iii) illustrates an oval-shaped basophil surrounded by some cellular substances. Despite the presence of these substances, since its cell membrane remains clearly distinguishable, we annotated as having a round cell shape. However, the Grad-CAM heatmap reveals that the substances outside the cell are interpreted as the cell boundary, leading to an incorrect prediction of an irregular cell shape for this basophil.

(iv) **Overlooking WBC**. We have observed instances where the WBC is overlooked, and the Grad-CAM heatmap highlights the red blood cell (RBC) instead, as depicted in Figure 13-(iv). Consequently, this leads to incorrect predictions, particularly when estimating cell size since RBCs generally have smaller sizes compared to WBCs.

## 7.7  Towards Broader Applicability

We established the attributes by analyzing five major types of WBCs derived from peripheral blood samples of healthy human individuals. Nevertheless, our attribute definitions are applicable in broader contexts. Moreover, we anticipate that the attribute predictor trained on our dataset will demonstrate generalizability to other domains in the majority of cases. However, formally exploring this aspect requires constructing datasets and conducting rigorous evaluations, which constitute future work. As a preliminary step, we conducted small-scale case studies to evaluate the applicability of the attribute predictor to cell images beyond our dataset. Specifically, we manually examined a small number of cell images of (i) peripheral blood samples from COVID-19 patients [29], (ii) bone marrow instead of peripheral blood [45], as well as (iii) peripheral blood samples from a non-human species, namely juvenile Visayan warty pigs [46].

(i) **Cells from COVID-19 Patients**. As shown in Figure 14-(i), the majority of predictions made by our model, which was trained on healthy peripheral blood cells, are consistent with the descriptions provided in [29] (listed at the bottom of respective image, in *italic*). However, there was one exception, as the prediction related to cytoplasm color in Figure 14-(i)(a) was inaccurately determined. This discrepancy could be attributed to the difference in staining compared to that of our training data.

(ii) **Cells from Bone Marrow**. As illustrated in Figure 14-(ii), our attribute predictor successfully identified the attributes that related to the cytoplasm and granules, showcasing its effectiveness in investigating blood cells in bone marrow samples. However, incorrect predictions were observed for cell and nucleus-related attributes, such as cell size and nucleus shape. The poor prediction

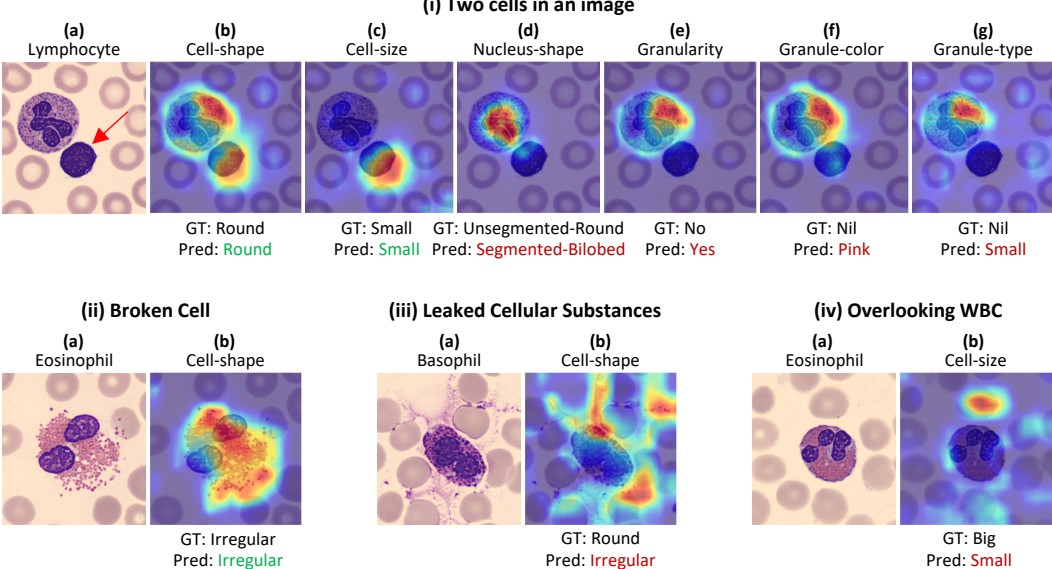

Figure 13: Potential factors contributing to incorrect attribute predictions. Please refer to Appendix 7.6 for more detailed descriptions. Correct predictions are highlighted in green, while incorrect predictions are highlighted in red.

performance in cell size may be due to the nature of the bone marrow dataset, which crops the WBCs at a higher magnification level. (To observe the difference in magnification levels, compare the red blood cells in Figure 14-(ii) to Figure 14-(i) and Figure 14-(iii).) Since we define the size of a cell based on its relative size compared to the red blood cells, all WBCs in Figure 14-(ii) are small cells, even though they occupy a larger number of pixels in the image. This could also mean that our model did not actually learn the size in the way we defined it; it may be simply checking the absolute size in the image rather than checking the size relative to the red blood cells. Exploring this further is future work.

(iii) **Cells from Pigs**. To investigate the applicability beyond human blood samples, we explored animal blood smears, as shown in Figure 14-(iii). We observe that their staining and smear preparation methods, which are different from the PBC dataset, have a substantial impact on the prediction performance, particularly for color-related attributes. In Figure 14-(iii)(b), for instance, the model fails to predict the presence of eosinophil granules due to significant differences compared to eosinophils in other datasets. (Please refer to Figure 14-(ii)(c) and Figure 13-(iv)(a) for the granule color of eosinophils in the bone marrow dataset and the PBC dataset, respectively.) Moreover, the nucleus and granules of Figure 14-(iii)(a) is hardly recognizable, even to human observers. This difficulty in recognition leads to incorrect predictions for nucleus shape.

## 7.8 Details of Attribute-based WBC Classifier in Section 5.1

We simply use the predicted probability (ranging from 0 to 1) of each attribute value to predict the cell types. This (lazy) formulation can cause the multicollinearity problem, so we employ the L1 regularizer. The train/val/test split is the same as our dataset split. The specific linear classifier we use is `sklearn.linear_model.SGDClassifier(max_iter=1000, loss="log_loss", penalty="l1")` in scikit-learn[1].

---

[1] `https://scikit-learn.org/stable/modules/generated/sklearn.linear_model.SGDClassifier.html`

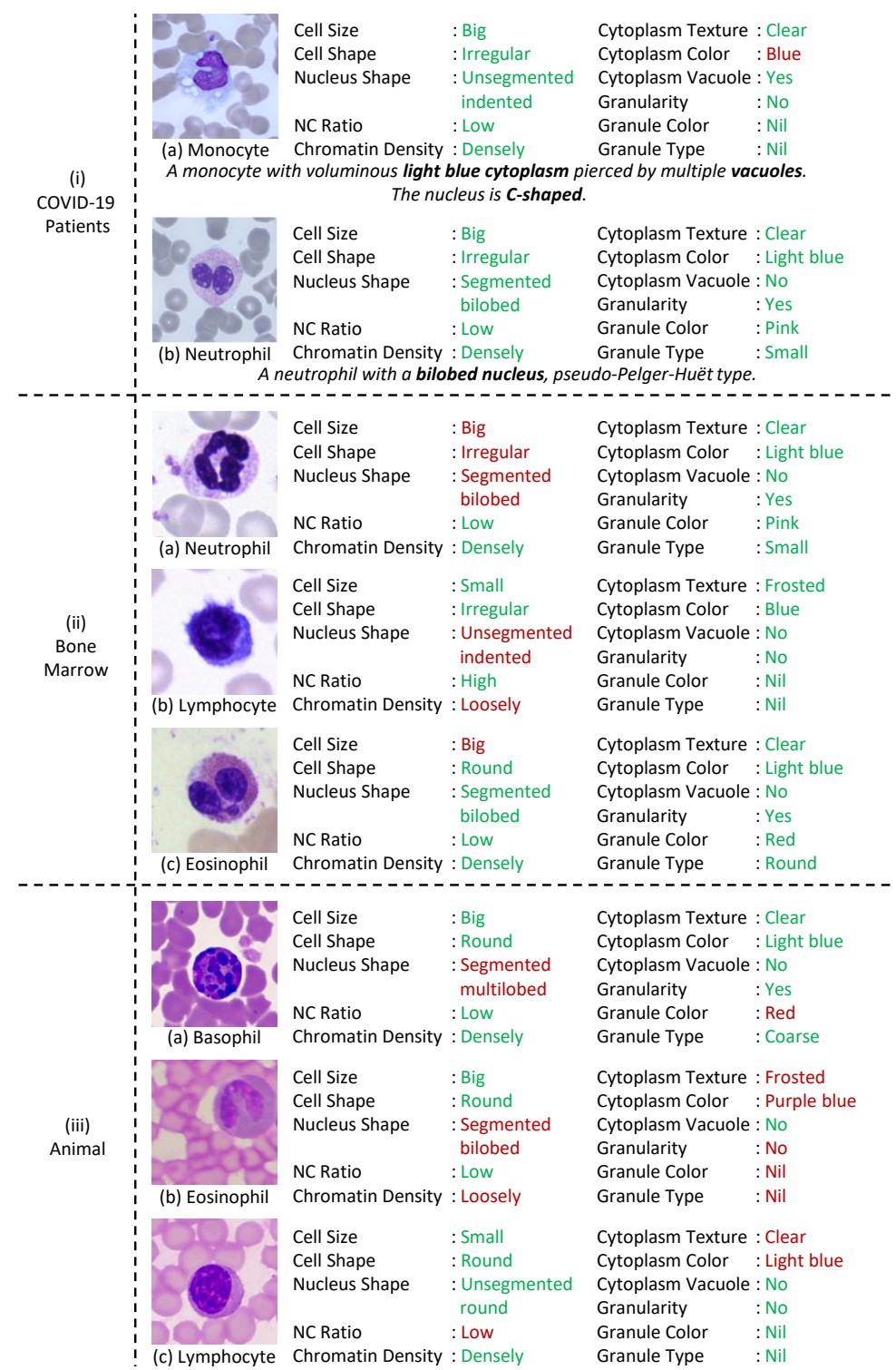

**(i) COVID-19 Patients**

**(a) Monocyte**

| | | | |
|---|---|---|---|
| Cell Size | : Big | Cytoplasm Texture | : Clear |
| Cell Shape | : Irregular | Cytoplasm Color | : Blue |
| Nucleus Shape | : Unsegmented indented | Cytoplasm Vacuole | : Yes |
| | | Granularity | : No |
| NC Ratio | : Low | Granule Color | : Nil |
| Chromatin Density | : Densely | Granule Type | : Nil |

*A monocyte with voluminous **light blue cytoplasm** pierced by multiple **vacuoles**. The nucleus is **C-shaped**.*

**(b) Neutrophil**

| | | | |
|---|---|---|---|
| Cell Size | : Big | Cytoplasm Texture | : Clear |
| Cell Shape | : Irregular | Cytoplasm Color | : Light blue |
| Nucleus Shape | : Segmented bilobed | Cytoplasm Vacuole | : No |
| | | Granularity | : Yes |
| NC Ratio | : Low | Granule Color | : Pink |
| Chromatin Density | : Densely | Granule Type | : Small |

*A neutrophil with a **bilobed nucleus**, pseudo-Pelger-Huët type.*

**(ii) Bone Marrow**

**(a) Neutrophil**

| | | | |
|---|---|---|---|
| Cell Size | : Big | Cytoplasm Texture | : Clear |
| Cell Shape | : Irregular | Cytoplasm Color | : Light blue |
| Nucleus Shape | : Segmented bilobed | Cytoplasm Vacuole | : No |
| | | Granularity | : Yes |
| NC Ratio | : Low | Granule Color | : Pink |
| Chromatin Density | : Densely | Granule Type | : Small |

**(b) Lymphocyte**

| | | | |
|---|---|---|---|
| Cell Size | : Small | Cytoplasm Texture | : Frosted |
| Cell Shape | : Irregular | Cytoplasm Color | : Blue |
| Nucleus Shape | : Unsegmented indented | Cytoplasm Vacuole | : No |
| | | Granularity | : No |
| NC Ratio | : High | Granule Color | : Nil |
| Chromatin Density | : Loosely | Granule Type | : Nil |

**(c) Eosinophil**

| | | | |
|---|---|---|---|
| Cell Size | : Big | Cytoplasm Texture | : Clear |
| Cell Shape | : Round | Cytoplasm Color | : Light blue |
| Nucleus Shape | : Segmented bilobed | Cytoplasm Vacuole | : No |
| | | Granularity | : Yes |
| NC Ratio | : Low | Granule Color | : Red |
| Chromatin Density | : Densely | Granule Type | : Round |

**(iii) Animal**

**(a) Basophil**

| | | | |
|---|---|---|---|
| Cell Size | : Big | Cytoplasm Texture | : Clear |
| Cell Shape | : Round | Cytoplasm Color | : Light blue |
| Nucleus Shape | : Segmented multilobed | Cytoplasm Vacuole | : No |
| | | Granularity | : Yes |
| NC Ratio | : Low | Granule Color | : Red |
| Chromatin Density | : Densely | Granule Type | : Coarse |

**(b) Eosinophil**

| | | | |
|---|---|---|---|
| Cell Size | : Big | Cytoplasm Texture | : Frosted |
| Cell Shape | : Round | Cytoplasm Color | : Purple blue |
| Nucleus Shape | : Segmented bilobed | Cytoplasm Vacuole | : No |
| | | Granularity | : No |
| NC Ratio | : Low | Granule Color | : Nil |
| Chromatin Density | : Loosely | Granule Type | : Nil |

**(c) Lymphocyte**

| | | | |
|---|---|---|---|
| Cell Size | : Small | Cytoplasm Texture | : Clear |
| Cell Shape | : Round | Cytoplasm Color | : Light blue |
| Nucleus Shape | : Unsegmented round | Cytoplasm Vacuole | : No |
| | | Granularity | : No |
| NC Ratio | : Low | Granule Color | : Nil |
| Chromatin Density | : Densely | Granule Type | : Nil |

Figure 14: The prediction results of our attribute predictor were evaluated for characterizing cells obtained from: (i) peripheral blood smears of patients with COVID-19, (ii) bone marrow smears, and (iii) peripheral blood smears of animal (juvenile Visayan warty pigs). The descriptions of (i)(a) and (i)(b) were summarized from [29]. Correct predictions are highlighted in green, while incorrect predictions are highlighted in red.

## 7.9 Details of StyleGAN-based Image Editing in Section 5.2

We trained StyleGAN-v2[2] [51] and subsequently pixel2style2pixel (pSp)[3] [68] encoder. These models enable us to embed cell images into the latent space of the trained StyleGAN, allowing for GAN inversion. Theoretically, this approach allows us to embed any cell image into the GAN latent space. By manipulating the latent space, we were able to edit the generated images, which is shown in the main paper. Specifically, we explored two techniques: GANSpace[4] [52], which applies Principal Components Analysis (PCA) to the latent space, and StyleSpace[5] [53], which can identify the subspace of the GAN latent space corresponding to certain attributes based on example images. Using these methods, we discovered that a specific principal component can control the cell size and NC ratio, as depicted in Figure 7 in the main paper. Additionally, although not presented in the main paper, we found subspaces to edit nucleus shape (Figure 15-(a)) and cytoplasm texture (Figure 15-(b)). We acknowledge that we did not rigorously evaluate controllability and image quality of these two techniques, as our primary aim was to showcase the usability of our attribute dataset. The code used for these experiments was obtained from the authors of the referenced papers (see the corresponding footnotes).

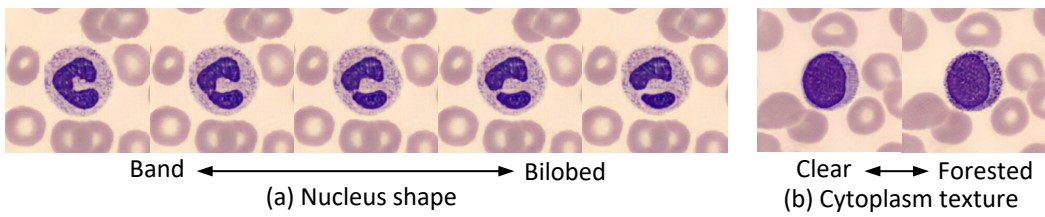

Figure 15: Applying StyleSpace [53] on our attribute annotation, we discovered the way to control the necleus shape and cytoplasm-texture.

## 7.10 Details of APL Detection in Section 5.3

We used the images provided by the dataset [59] and followed their proposed data split except that we removed the duplicated and corrupted images that we identified. We publicly reported the duplicates[6] and the corruptions[7], but we have not received any comments thus far. Once ResNet50 was trained to classify APL or non-APL, we applied our attribute predictors to the promyelocyte images in the test set. Subsequently, we visualized the probability distribution of each attribute for the four groups (true positive, false negative, true negative, and false positive). Upon manual investigation of all attributes, we discovered a significant bias in the blue cytoplasm, as illustrated in Figure 8-(a)(b) in the main paper. It is worth noting that the inherent morphological differences between APL and non-APL promyelocytes may lead to distinct attribute distributions. In other words, if an attribute is a crucial feature for diagnosing APL, the bias is not necessarily undesirable. However, in this specific case, we found that the correlation only existed in the training set, and we could not find any literature supporting this bias. Consequently, we concluded that the correlation was spurious.

## 7.11 Attribute-based Acute Lymphocytic Leukemia Prediction

Our dataset provides annotations solely for images of healthy patients. However, we anticipate that the attributes we've identified can be exploited for downstream tasks with direct clinical implications, such as certain disorders. In this context, we investigate the potential use of these attributes for the detection of Acute Lymphocytic Leukemia (ALL), which is a significant challenge in the field of pathology. We employ the ALL-IDB [4] dataset, specifically the ALL-IDB2 subset, which includes cropped cell images. We follow the train/val/test split and baselines reported by Genovese et al. [69], who are from the same research group that developed the ALL-IDB dataset.

---

[2]https://github.com/NVlabs/stylegan3

[3]https://github.com/eladrich/pixel2style2pixel

[4]https://github.com/harskish/ganspace

[5]https://github.com/betterze/StyleSpace

[6]https://github.com/sidhomj/DeepAPL/issues/32

[7]https://github.com/sidhomj/DeepAPL/issues/31

Table 2: Results for Predicting Acute Lymphocytic Leukemia (ALL) on the ALL-IDB [4].

| Method | Accuracy (%) |
|---|---|
| VGG16 | $87.54 \pm 3.15$ [69] |
| ResNet18 | $88.69 \pm 2.67$ [69] |
| ResNet18 Pre-trained on Additional Histopathological Images | $97.92 \pm 1.62$ [69] |
| ViT-Base | $\mathbf{91.28} \pm 2.17$ |
| Logistic Regression on Predicted Attributes | $81.03 \pm 1.48$ |
| Gradient Boosting on Predicted Attributes | $83.85 \pm 1.88$ |
| Logistic Regression on Attributes Predicted with Domain Adaptation | $86.92 \pm 2.13$ |
| Gradient Boosting on Attributes Predicted with Domain Adaptation | $\mathbf{90.77} \pm 1.23$ |

Due to the lack of attribute annotations for the ALL-IDB dataset, we employ our model, trained on our dataset, to predict attributes. We utilize the ViT-Base backbone for this task, as our preliminary examination of a small number of cases indicates that the predicted attributes are more accurate than those from other backbones. This observation aligns with previous research [70], which suggests that ViTs typically yield higher cross-dataset accuracy than CNNs. We also implement a straightforward domain adaptation technique to concurrently minimize the classification loss and the Maximum Mean Discrepancy (MMD) [71] loss between the labeled domain (our dataset) and the unlabeled test domain (the ALL-IDB dataset).

After obtaining the predicted attributes on ALL-IDB images, we trained two types of binary classifiers for ALL detection: L1-regularized multiclass logistic regression[8] and a gradient boosting classifier[9]. This is done using the predicted attributes with and without domain adaptation. We ran the code three times with three different seeds and reported the mean and 95% confidence intervals. We compared these results with the image-based classifiers using ViT-Base, as well as ResNet18 and VGG16 (the two baselines reported in [69]).

Table 2 summarizes the results. The accuracies of attribute-based prediction methods are comparable to those of image-based prediction methods, suggesting the feasibility of using our attributes as interpretable features for clinical downstream tasks.

### 7.12 Peer Loss for Potential Annotation Noise

In response to a suggestion from a reviewer, we have explored the peer loss [72], which is robust against label noise and does not require to specify the noise rate. We conducted the following experiments using the ResNet50 backbone and executed the code three times to obtain the mean and the 95% confidence intervals.

Table 3: Impact of Peer Loss on Average Macro F1 Score

| Dataset | Utilize Peer Loss? | Average Macro F1 Score (%) |
|---|---|---|
| Training Set with 10% Label Noise | No | $85.31 \pm 0.24$ |
| Training Set with 10% Label Noise | Yes | $88.86 \pm 0.02$ |
| Training Set without Added Label Noise | No | $91.20 \pm 0.06$ |
| Training Set without Added Label Noise | Yes | $91.17 \pm 0.03$ |

To assess the efficacy of the peer loss on our dataset, we intentionally introduced noise by perturbing 10% of the annotations within the training data. Specifically, from the pool of 67,969 categorical values (6,179 images $\times$ 11 attributes), we randomly selected 6,797 values (10%) and replaced each with an alternative value chosen uniformly from the remaining possibilities. Using this perturbed data, we trained our model both with and without incorporating the peer loss. The outcomes, presented

---

[8]`https://scikit-learn.org/stable/modules/generated/sklearn.linear_model.`
`SGDClassifier.html`

[9]`https://scikit-learn.org/stable/modules/generated/sklearn.ensemble.`
`GradientBoostingClassifier.html`

in the upper section of Table 3, reveal that training on the noisy data alone yields an average macro F1 score of $85.31 \pm 0.24$. However, integrating the peer loss increases this score to $88.86 \pm 0.02$, demonstrating the efficacy of the peer loss.

After confirming the effectiveness of peer loss, we applied it to our training data without any artificially introduced noise. As shown in the lower part of Table 3, utilizing peer loss yields an average macro F1 score of $91.17 \pm 0.03$, while training without it results in a score of $91.20 \pm 0.06$. The nearly identical scores indicate that the impact of peer loss is not significant, suggesting that the presence of annotation noise within our dataset is limited.

### 7.13 Other Tables and Figures

- Table 4: Frequencies of attribute values, illustrating content similar to Figure 3 in the main paper.
- Table 5: Initial attributes used as a starting point to differentiate the five types of WBCs.
- Table 7: Precision, recall, and F-measure values for each attribute value.
- Figure 16: GradCAM heatmaps for different seeds and backbones that are not reported in the main paper.
- Figure 17: Visualization of how staining conditions and imaging equipment influence the attributes.

Table 4: Attribute Dist. The distribution represents the results of annotating all typical WBCs from the PBC dataset, which is the image source we utilized. We did not actively control or manipulate the distribution.

| Attribute | Value (Count) |
|---|---|
| Cell-Size | Big (4,997), Small (4,271) |
| Cell-Shape | Round (7,173), Irregular (2,095) |
| Nucleus-Shape | Segmented-Bilobed (2,806), Unsegmented-Band (2,356), Unsegmented-Indented (1,205), Segmented-Multilobed (1,143), Unsegmented-Round (967), Irregular (791) |
| Nuclear-Cytoplasmic-Ratio | Low (8,148), High (1,120) |
| Chromatin-Density | Densely (8,443), Loosely (825) |
| Cytoplasm-Vacuole | No (8,559), Yes (709) |
| Cytoplasm-Texture | Clear (7,429), Frosted (1,839) |
| Cytoplasm-Color | Light Blue (7,011), Blue (1,273), Purple Blue (984) |
| Granule-Type | Small (3,003), Round (2,801), Nil (2,374), Coarse (1,090) |
| Granule-Color | Pink (2,925), Red (2,803), Nil (2,373), Purple (1,167) |
| Granularity | Yes (6,896), No (2,372) |

Table 5: Coarse Morphological Attributes

| Cell Type | Nucleus Structure | NC Ratio | Granularity | Granularity Color | Cell Size |
|---|---|---|---|---|---|
| Basophils | Segmented | Low | Yes | Blue / Black (dense) | |
| Eosinophils | Segmented | Low | Yes | Red | |
| Lymphocytes | Unsegmented | High | No | | Small |
| Monocytes | Unsegmented | Low | No | | |
| Neutrophils | Segmented | Low | Yes | Blue | |

Table 6: Macro Precision (Pre.), Macro Recall (Rec.), and Macro F-measure (F-m.) of VGG16, ResNet50, ViT-Base, and ConvNeXt-Tiny (CNXT-T) for the task of attribute prediction.

| | Cell Size | Cell Shape | Nucleus Shape | Nuclear Cyto-plasmic Ratio | Chromatin Density | Cytoplasm Vacuole | Cytoplasm Texture | Cytoplasm Color | Granule Type | Granule Color | Granularity | (Average) |
|---|---|---|---|---|---|---|---|---|---|---|---|---|
| VGG16 Pre. | 83.45 ±0.61 | 89.08 ±1.80 | 74.88 ±0.81 | 96.78 ±1.01 | 83.97 ±1.26 | 91.24 ±0.83 | 92.29 ±0.47 | 84.19 ±0.32 | 99.28 ±0.10 | 98.72 ±0.31 | 99.57 ±0.09 | 90.31 ±0.51 |
| VGG16 Rec. | 83.44 ±0.43 | 90.13 ±0.67 | 74.18 ±0.50 | 95.12 ±0.94 | 86.71 ±2.56 | 85.96 ±1.39 | 95.21 ±1.38 | 83.93 ±0.96 | 99.42 ±0.12 | 98.54 ±0.11 | 99.60 ±0.05 | 90.20 ±0.70 |
| VGG16 F-m. | 83.44 ±0.52 | 89.52 ±0.78 | 74.10 ±0.39 | 95.91 ±0.35 | 85.18 ±0.74 | 88.36 ±0.63 | 93.61 ±0.35 | 83.95 ±0.27 | 99.35 ±0.10 | 98.62 ±0.18 | 99.58 ±0.07 | 90.15 ±0.31 |
| ResNet50 Pre. | 84.21 ±0.61 | 90.73 ±0.90 | 77.08 ±0.47 | 97.47 ±0.18 | 84.55 ±0.48 | 92.71 ±1.76 | 93.22 ±0.87 | 88.24 ±0.30 | 99.36 ±0.10 | 98.89 ±0.08 | 99.60 ±0.01 | 91.46 ±0.30 |
| ResNet50 Rec. | 83.69 ±0.44 | 90.64 ±0.80 | 75.70 ±0.99 | 95.30 ±0.05 | 88.52 ±0.59 | 87.08 ±2.09 | 95.95 ±0.42 | 88.06 ±0.45 | 99.52 ±0.10 | 98.63 ±0.10 | 99.62 ±0.03 | 91.16 ±0.26 |
| ResNet50 F-m. | 83.81 ±0.33 | 90.66 ±0.36 | 76.13 ±0.59 | 96.35 ±0.06 | 86.39 ±0.32 | 89.57 ±0.47 | 94.49 ±0.51 | 87.99 ±0.47 | 99.44 ±0.07 | 98.76 ±0.08 | 99.61 ±0.02 | 91.20 ±0.06 |
| ViT-B Pre. | 83.58 ±0.14 | 89.53 ±0.64 | 77.21 ±0.27 | 96.87 ±0.75 | 84.73 ±0.97 | 90.62 ±2.74 | 92.86 ±0.16 | 87.62 ±0.18 | 99.32 ±0.13 | 99.16 ±0.06 | 99.64 ±0.02 | 91.01 ±0.30 |
| ViT-B Rec. | 83.01 ±0.52 | 90.08 ±0.52 | 75.88 ±2.03 | 95.91 ±0.76 | 84.61 ±2.93 | 90.68 ±0.70 | 95.18 ±0.96 | 88.04 ±0.60 | 99.45 ±0.10 | 98.70 ±0.19 | 99.66 ±0.05 | 91.02 ±0.32 |
| ViT-B F-m. | 83.19 ±0.43 | 89.80 ±0.51 | 75.94 ±1.24 | 96.37 ±0.29 | 84.51 ±1.78 | 90.63 ±0.19 | 93.94 ±0.38 | 87.70 ±0.45 | 99.39 ±0.11 | 98.92 ±0.11 | 99.65 ±0.02 | 90.91 ±0.21 |
| CNXT-T Pre. | 83.50 ±0.43 | 91.00 ±0.61 | 78.51 ±1.27 | 96.80 ±0.42 | 85.56 ±0.68 | 92.84 ±0.61 | 93.93 ±0.54 | 88.05 ±0.19 | 99.49 ±0.09 | 99.22 ±0.16 | 99.70 ±0.06 | 91.69 ±0.26 |
| CNXT-T Rec. | 83.44 ±0.35 | 91.54 ±0.39 | 77.68 ±1.89 | 95.65 ±0.29 | 85.87 ±0.85 | 88.04 ±0.26 | 95.00 ±0.77 | 88.28 ±0.77 | 99.64 ±0.04 | 98.75 ±0.08 | 99.71 ±0.03 | 91.24 ±0.26 |
| CNXT-T F-m. | 83.46 ±0.37 | 91.26 ±0.50 | 77.97 ±1.58 | 96.21 ±0.34 | 85.70 ±0.31 | 90.26 ±0.40 | 94.44 ±0.27 | 88.15 ±0.46 | 99.56 ±0.06 | 98.98 ±0.12 | 99.71 ±0.04 | 91.43 ±0.26 |

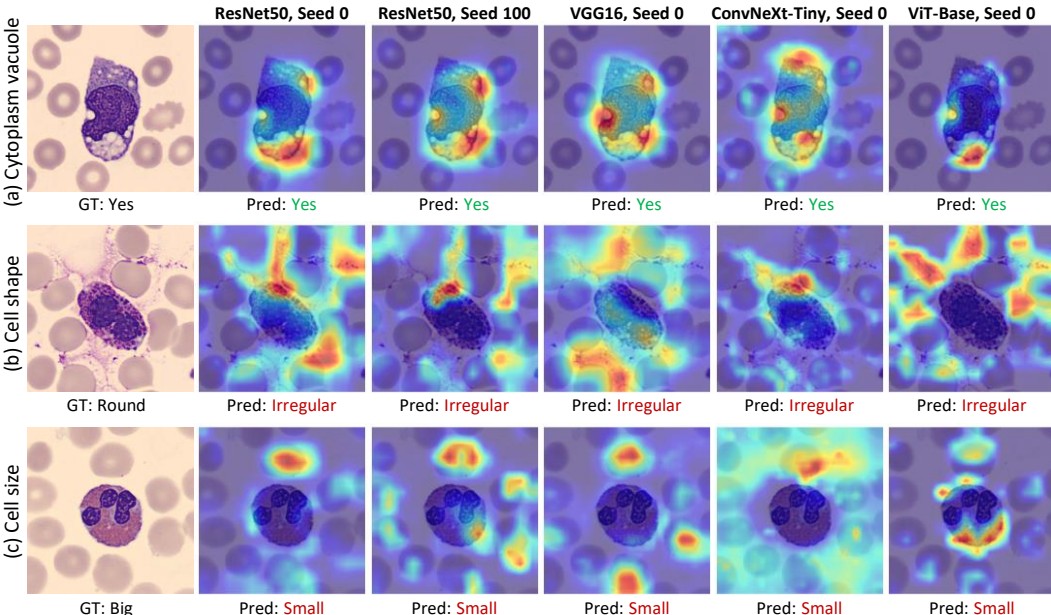

Figure 16: Grad-CAM heatmaps from different seeds (ResNet50 with seed 100) and various backbones (including VGG16, ConvNeXt-Tiny, and Vit-Base), while images are the same as Figure 4

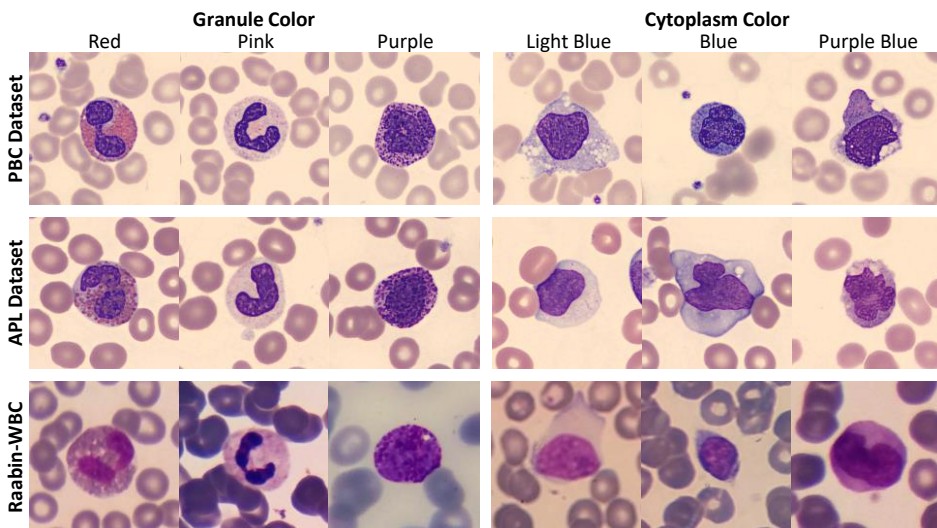

Figure 17: WBC appearances under different staining and microscopy conditions. WBCs sourced from the PBC Dataset [6] (used in this work) have undergone May Grünwald-Giemsa staining and were acquired through the automated digital cell morphology analyzer, CellaVision DM96. WBCs from the APL Dataset [59] are stained using Wright's method and observed through the CellaVision DM100 cell analyzer. On the other hand, the Raabin-WBC [7] images, stained with Giemsa, are captured using smartphones mounted on both Olympus CX18 and Zeiss microscopes.

Table 7: Precision, Recall, and F-measure of VGG16, ResNet50, ViT-Base, and ConvNeXt-Tiny (CNXT-T) for Each Attribute Value

| Backbone | Attribute Name | Attribute Value | Precision | Recall | F-measure |
|---|---|---|---|---|---|
| VGG16 | Cell Size | Big | 85.17 ± 0.32 | 84.98 ± 1.65 | 85.06 ± 0.67 |
| VGG16 | Cell Size | Small | 81.73 ± 1.52 | 81.91 ± 0.80 | 81.81 ± 0.37 |
| VGG16 | Cell Shape | Irregular | 82.37 ± 4.34 | 85.71 ± 2.99 | 83.89 ± 1.01 |
| VGG16 | Cell Shape | Round | 95.79 ± 0.77 | 94.54 ± 1.82 | 95.15 ± 0.57 |
| VGG16 | Nucleus Shape | Irregular | 56.35 ± 2.93 | 67.55 ± 7.88 | 61.13 ± 2.08 |
| VGG16 | Nucleus Shape | Segmented-Bilobed | 74.98 ± 0.65 | 74.42 ± 0.73 | 74.70 ± 0.17 |
| VGG16 | Nucleus Shape | Segmented-Multilobed | 72.13 ± 4.22 | 55.82 ± 4.45 | 62.72 ± 1.69 |
| VGG16 | Nucleus Shape | Unsegmented-Band | 80.65 ± 0.78 | 88.86 ± 0.42 | 84.55 ± 0.29 |
| VGG16 | Nucleus Shape | Unsegmented-Indented | 81.87 ± 0.61 | 76.22 ± 3.09 | 78.91 ± 1.55 |
| VGG16 | Nucleus Shape | Unsegmented-Round | 83.31 ± 3.21 | 82.19 ± 5.07 | 82.60 ± 1.51 |
| VGG16 | NC Ratio | High | 94.80 ± 2.25 | 90.93 ± 2.15 | 92.79 ± 0.61 |
| VGG16 | NC Ratio | Low | 98.76 ± 0.29 | 99.30 ± 0.32 | 99.03 ± 0.09 |
| VGG16 | Chromatin Density | Densely | 97.61 ± 0.57 | 96.66 ± 0.68 | 97.13 ± 0.12 |
| VGG16 | Chromatin Density | Loosely | 70.34 ± 2.98 | 76.77 ± 5.76 | 73.23 ± 1.45 |
| VGG16 | Cytoplasm Vacuole | No | 97.72 ± 0.24 | 98.87 ± 0.20 | 98.29 ± 0.07 |
| VGG16 | Cytoplasm Vacuole | Yes | 84.77 ± 1.83 | 73.06 ± 2.95 | 78.43 ± 1.20 |
| VGG16 | Cytoplasm Texture | Clear | 98.52 ± 0.91 | 95.99 ± 0.75 | 97.23 ± 0.07 |
| VGG16 | Cytoplasm Texture | Frosted | 86.05 ± 1.84 | 94.43 ± 3.51 | 89.98 ± 0.63 |
| VGG16 | Cytoplasm Color | Blue | 80.39 ± 2.33 | 77.48 ± 3.15 | 78.83 ± 0.60 |
| VGG16 | Cytoplasm Color | Light Blue | 98.90 ± 0.61 | 99.11 ± 0.26 | 99.00 ± 0.20 |
| VGG16 | Cytoplasm Color | Purple Blue | 73.27 ± 2.98 | 75.20 ± 6.06 | 74.01 ± 1.30 |
| VGG16 | Granule Type | Coarse | 98.76 ± 0.15 | 99.52 ± 0.31 | 99.14 ± 0.13 |
| VGG16 | Granule Type | Nil | 99.22 ± 0.07 | 99.51 ± 0.11 | 99.36 ± 0.09 |
| VGG16 | Granule Type | Round | 99.75 ± 0.06 | 99.46 ± 0.30 | 99.60 ± 0.13 |
| VGG16 | Granule Type | Small | 99.41 ± 0.33 | 99.18 ± 0.11 | 99.29 ± 0.16 |
| VGG16 | Granule Color | Nil | 99.18 ± 0.07 | 99.63 ± 0.00 | 99.40 ± 0.03 |
| VGG16 | Granule Color | Pink | 98.46 ± 0.40 | 98.92 ± 0.60 | 98.69 ± 0.14 |
| VGG16 | Granule Color | Purple | 97.40 ± 1.49 | 96.17 ± 0.62 | 96.77 ± 0.74 |
| VGG16 | Granule Color | Red | 99.82 ± 0.11 | 99.42 ± 0.21 | 99.62 ± 0.05 |
| VGG16 | Granularity | No | 99.34 ± 0.17 | 99.42 ± 0.07 | 99.38 ± 0.10 |
| VGG16 | Granularity | Yes | 99.80 ± 0.02 | 99.77 ± 0.06 | 99.78 ± 0.04 |
| ResNet50 | Cell Size | Big | 84.23 ± 2.02 | 87.54 ± 3.27 | 85.79 ± 0.56 |
| ResNet50 | Cell Size | Small | 84.18 ± 3.06 | 79.83 ± 3.89 | 81.84 ± 0.78 |
| ResNet50 | Cell Shape | Irregular | 85.70 ± 2.35 | 85.48 ± 2.40 | 85.54 ± 0.55 |
| ResNet50 | Cell Shape | Round | 95.77 ± 0.63 | 95.80 ± 0.91 | 95.78 ± 0.22 |
| ResNet50 | Nucleus Shape | Irregular | 62.24 ± 0.70 | 62.37 ± 5.10 | 62.21 ± 2.51 |
| ResNet50 | Nucleus Shape | Segmented-Bilobed | 74.07 ± 4.38 | 80.30 ± 4.59 | 76.86 ± 0.65 |
| ResNet50 | Nucleus Shape | Segmented-Multilobed | 71.43 ± 3.96 | 59.97 ± 6.11 | 64.95 ± 3.25 |
| ResNet50 | Nucleus Shape | Unsegmented-Band | 85.42 ± 2.87 | 85.05 ± 4.68 | 85.10 ± 1.13 |
| ResNet50 | Nucleus Shape | Unsegmented-Indented | 85.00 ± 3.44 | 80.50 ± 3.33 | 82.58 ± 0.21 |
| ResNet50 | Nucleus Shape | Unsegmented-Round | 84.35 ± 2.91 | 86.01 ± 3.79 | 85.07 ± 0.39 |
| ResNet50 | NC Ratio | High | 96.16 ± 0.38 | 91.11 ± 0.14 | 93.56 ± 0.11 |
| ResNet50 | NC Ratio | Low | 98.79 ± 0.02 | 99.50 ± 0.05 | 99.14 ± 0.02 |
| ResNet50 | Chromatin Density | Densely | 97.97 ± 0.13 | 96.67 ± 0.20 | 97.31 ± 0.07 |
| ResNet50 | Chromatin Density | Loosely | 71.14 ± 1.02 | 80.37 ± 1.30 | 75.46 ± 0.58 |
| ResNet50 | Cytoplasm Vacuole | No | 97.89 ± 0.37 | 99.05 ± 0.39 | 98.47 ± 0.01 |
| ResNet50 | Cytoplasm Vacuole | Yes | 87.53 ± 3.90 | 75.10 ± 4.57 | 80.67 ± 0.94 |
| ResNet50 | Cytoplasm Texture | Clear | 98.78 ± 0.30 | 96.49 ± 0.64 | 97.62 ± 0.25 |
| ResNet50 | Cytoplasm Texture | Frosted | 87.66 ± 1.92 | 95.42 ± 1.16 | 91.36 ± 0.77 |
| ResNet50 | Cytoplasm Color | Blue | 83.30 ± 3.68 | 87.53 ± 4.93 | 85.19 ± 0.49 |
| ResNet50 | Cytoplasm Color | Light Blue | 99.37 ± 0.32 | 99.09 ± 0.25 | 99.23 ± 0.09 |
| ResNet50 | Cytoplasm Color | Purple Blue | 82.07 ± 3.01 | 77.55 ± 6.01 | 79.54 ± 1.76 |
| ResNet50 | Granule Type | Coarse | 98.67 ± 0.54 | 99.52 ± 0.31 | 99.09 ± 0.20 |
| ResNet50 | Granule Type | Nil | 99.22 ± 0.28 | 99.63 ± 0.11 | 99.43 ± 0.18 |
| ResNet50 | Granule Type | Round | 99.82 ± 0.11 | 99.89 ± 0.10 | 99.86 ± 0.03 |
| ResNet50 | Granule Type | Small | 99.74 ± 0.11 | 99.05 ± 0.19 | 99.39 ± 0.10 |
| ResNet50 | Granule Color | Nil | 99.34 ± 0.07 | 99.71 ± 0.13 | 99.53 ± 0.09 |
| ResNet50 | Granule Color | Pink | 98.66 ± 0.28 | 98.89 ± 0.25 | 98.77 ± 0.07 |
| ResNet50 | Granule Color | Purple | 97.82 ± 0.25 | 95.99 ± 0.43 | 96.90 ± 0.25 |
| ResNet50 | Granule Color | Red | 99.75 ± 0.25 | 99.93 ± 0.12 | 99.84 ± 0.09 |
| ResNet50 | Granularity | No | 99.38 ± 0.00 | 99.46 ± 0.07 | 99.42 ± 0.03 |

Continued on next page

Table 7: Precision, Recall, and F-measure of VGG16, ResNet50, ViT-Base, and ConvNeXt-Tiny (CNXT-T) for Each Attribute Value

| Backbone | Attribute Name | Attribute Value | Precision | Recall | F-measure |
|---|---|---|---|---|---|
| ResNet50 | Granularity | Yes | 99.81 ± 0.02 | 99.78 ± 0.00 | 99.80 ± 0.01 |
| ViT-Base | Cell Size | Big | 83.20 ± 1.36 | 87.72 ± 1.49 | 85.38 ± 0.13 |
| ViT-Base | Cell Size | Small | 83.96 ± 1.22 | 78.30 ± 2.50 | 81.00 ± 0.80 |
| ViT-Base | Cell Shape | Irregular | 83.44 ± 1.25 | 85.10 ± 1.06 | 84.25 ± 0.78 |
| ViT-Base | Cell Shape | Round | 95.63 ± 0.29 | 95.07 ± 0.45 | 95.35 ± 0.25 |
| ViT-Base | Nucleus Shape | Irregular | 68.62 ± 5.78 | 64.77 ± 18.26 | 64.79 ± 8.39 |
| ViT-Base | Nucleus Shape | Segmented-Bilobed | 73.59 ± 5.70 | 76.88 ± 6.31 | 74.83 ± 1.14 |
| ViT-Base | Nucleus Shape | Segmented-Multilobed | 69.59 ± 2.87 | 59.41 ± 5.31 | 63.89 ± 1.95 |
| ViT-Base | Nucleus Shape | Unsegmented-Band | 82.96 ± 5.54 | 84.48 ± 7.77 | 83.31 ± 1.72 |
| ViT-Base | Nucleus Shape | Unsegmented-Indented | 85.63 ± 1.94 | 80.96 ± 2.99 | 83.17 ± 0.65 |
| ViT-Base | Nucleus Shape | Unsegmented-Round | 82.85 ± 2.36 | 88.74 ± 2.35 | 85.66 ± 1.14 |
| ViT-Base | NC Ratio | High | 94.77 ± 1.69 | 92.53 ± 1.72 | 93.61 ± 0.52 |
| ViT-Base | NC Ratio | Low | 98.98 ± 0.23 | 99.29 ± 0.25 | 99.13 ± 0.07 |
| ViT-Base | Chromatin Density | Densely | 97.16 ± 0.63 | 97.10 ± 1.04 | 97.12 ± 0.39 |
| ViT-Base | Chromatin Density | Loosely | 72.29 ± 5.74 | 72.13 ± 6.53 | 71.90 ± 3.25 |
| ViT-Base | Cytoplasm Vacuole | No | 98.53 ± 0.14 | 98.51 ± 0.25 | 98.52 ± 0.06 |
| ViT-Base | Cytoplasm Vacuole | Yes | 82.70 ± 2.07 | 82.86 ± 1.64 | 82.75 ± 0.33 |
| ViT-Base | Cytoplasm Texture | Clear | 98.39 ± 0.58 | 96.45 ± 0.35 | 97.41 ± 0.12 |
| ViT-Base | Cytoplasm Texture | Frosted | 87.33 ± 0.84 | 93.91 ± 2.26 | 90.48 ± 0.64 |
| ViT-Base | Cytoplasm Color | Blue | 86.32 ± 1.59 | 80.50 ± 4.13 | 83.23 ± 1.46 |
| ViT-Base | Cytoplasm Color | Light Blue | 99.32 ± 0.40 | 99.21 ± 0.13 | 99.27 ± 0.14 |
| ViT-Base | Cytoplasm Color | Purple Blue | 77.22 ± 1.70 | 84.41 ± 2.59 | 80.61 ± 0.34 |
| ViT-Base | Granule Type | Coarse | 98.57 ± 0.52 | 99.33 ± 0.31 | 98.95 ± 0.30 |
| ViT-Base | Granule Type | Nil | 99.22 ± 0.28 | 99.38 ± 0.11 | 99.30 ± 0.09 |
| ViT-Base | Granule Type | Round | 99.89 ± 0.00 | 99.82 ± 0.15 | 99.86 ± 0.08 |
| ViT-Base | Granule Type | Small | 99.61 ± 0.09 | 99.28 ± 0.05 | 99.44 ± 0.07 |
| ViT-Base | Granule Color | Nil | 99.46 ± 0.07 | 99.55 ± 0.07 | 99.51 ± 0.00 |
| ViT-Base | Granule Color | Pink | 98.40 ± 0.40 | 99.49 ± 0.19 | 98.94 ± 0.14 |
| ViT-Base | Granule Color | Purple | 98.81 ± 0.37 | 95.81 ± 0.79 | 97.28 ± 0.26 |
| ViT-Base | Granule Color | Red | 99.96 ± 0.06 | 99.93 ± 0.06 | 99.95 ± 0.05 |
| ViT-Base | Granularity | No | 99.46 ± 0.06 | 99.51 ± 0.11 | 99.49 ± 0.03 |
| ViT-Base | Granularity | Yes | 99.83 ± 0.04 | 99.81 ± 0.02 | 99.82 ± 0.01 |
| CNXT-T | Cell Size | Big | 84.92 ± 0.64 | 85.43 ± 1.29 | 85.17 ± 0.45 |
| CNXT-T | Cell Size | Small | 82.09 ± 1.14 | 81.46 ± 1.16 | 81.76 ± 0.37 |
| CNXT-T | Cell Shape | Irregular | 85.71 ± 1.11 | 87.33 ± 0.46 | 86.51 ± 0.75 |
| CNXT-T | Cell Shape | Round | 96.28 ± 0.14 | 95.75 ± 0.37 | 96.01 ± 0.25 |
| CNXT-T | Nucleus Shape | Irregular | 63.15 ± 2.42 | 63.38 ± 7.48 | 63.15 ± 4.57 |
| CNXT-T | Nucleus Shape | Segmented-Bilobed | 75.91 ± 2.21 | 81.92 ± 1.84 | 78.78 ± 1.56 |
| CNXT-T | Nucleus Shape | Segmented-Multilobed | 72.04 ± 1.52 | 64.51 ± 4.39 | 68.01 ± 2.76 |
| CNXT-T | Nucleus Shape | Unsegmented-Band | 87.13 ± 2.24 | 85.48 ± 1.02 | 86.27 ± 0.64 |
| CNXT-T | Nucleus Shape | Unsegmented-Indented | 87.44 ± 1.94 | 81.57 ± 1.36 | 84.38 ± 0.47 |
| CNXT-T | Nucleus Shape | Unsegmented-Round | 85.38 ± 1.29 | 89.18 ± 0.80 | 87.23 ± 0.50 |
| CNXT-T | NC Ratio | High | 94.70 ± 0.79 | 92.00 ± 0.49 | 93.33 ± 0.61 |
| CNXT-T | NC Ratio | Low | 98.90 ± 0.07 | 99.29 ± 0.11 | 99.10 ± 0.08 |
| CNXT-T | Chromatin Density | Densely | 97.39 ± 0.19 | 97.29 ± 0.28 | 97.34 ± 0.07 |
| CNXT-T | Chromatin Density | Loosely | 73.73 ± 1.50 | 74.45 ± 1.94 | 74.06 ± 0.58 |
| CNXT-T | Cytoplasm Vacuole | No | 98.05 ± 0.04 | 99.07 ± 0.10 | 98.55 ± 0.06 |
| CNXT-T | Cytoplasm Vacuole | Yes | 87.62 ± 1.19 | 77.01 ± 0.44 | 81.97 ± 0.73 |
| CNXT-T | Cytoplasm Texture | Clear | 98.10 ± 0.50 | 97.23 ± 0.49 | 97.66 ± 0.10 |
| CNXT-T | Cytoplasm Texture | Frosted | 89.76 ± 1.49 | 92.76 ± 1.97 | 91.21 ± 0.44 |
| CNXT-T | Cytoplasm Color | Blue | 85.26 ± 0.63 | 83.90 ± 1.58 | 84.57 ± 0.69 |
| CNXT-T | Cytoplasm Color | Light Blue | 99.28 ± 0.36 | 99.18 ± 0.07 | 99.23 ± 0.15 |
| CNXT-T | Cytoplasm Color | Purple Blue | 79.60 ± 0.80 | 81.76 ± 2.32 | 80.65 ± 0.81 |
| CNXT-T | Granule Type | Coarse | 98.76 ± 0.40 | 99.71 ± 0.00 | 99.24 ± 0.20 |
| CNXT-T | Granule Type | Nil | 99.59 ± 0.07 | 99.55 ± 0.07 | 99.57 ± 0.06 |
| CNXT-T | Granule Type | Round | 99.89 ± 0.06 | 99.96 ± 0.06 | 99.93 ± 0.03 |
| CNXT-T | Granule Type | Small | 99.70 ± 0.09 | 99.34 ± 0.14 | 99.52 ± 0.07 |
| CNXT-T | Granule Color | Nil | 99.63 ± 0.00 | 99.59 ± 0.13 | 99.61 ± 0.07 |
| CNXT-T | Granule Color | Pink | 98.43 ± 0.05 | 99.60 ± 0.19 | 99.01 ± 0.10 |
| CNXT-T | Granule Color | Purple | 98.81 ± 0.63 | 95.81 ± 0.14 | 97.29 ± 0.37 |
| CNXT-T | Granule Color | Red | 100.00 ± 0.00 | 100.00 ± 0.00 | 100.00 ± 0.00 |
| CNXT-T | Granularity | No | 99.55 ± 0.13 | 99.59 ± 0.07 | 99.57 ± 0.06 |
| CNXT-T | Granularity | Yes | 99.85 ± 0.02 | 99.84 ± 0.05 | 99.85 ± 0.02 |

## Supplementary References

[62] Kaiming He, Xiangyu Zhang, Shaoqing Ren, and Jian Sun. Deep Residual Learning for Image Recognition. In *IEEE Conference on Computer Vision and Pattern Recognition*, 2016.

[63] Bec Crew. Google Scholar reveals its most influential papers for 2019. *Nature Index*, 2019.

[64] Karen Simonyan and Andrew Zisserman. Very deep convolutional networks for large-scale image recognition. In *International Conference on Learning Representations*, 2014.

[65] Alexey Dosovitskiy, Lucas Beyer, Alexander Kolesnikov, Dirk Weissenborn, Xiaohua Zhai, Thomas Unterthiner, Mostafa Dehghani, Matthias Minderer, Georg Heigold, Sylvain Gelly, Jakob Uszkoreit, and Neil Houlsby. An image is worth 16x16 words: Transformers for image recognition at scale. In *International Conference on Learning Representations*, 2021.

[66] Zhuang Liu, Hanzi Mao, Chao-Yuan Wu, Christoph Feichtenhofer, Trevor Darrell, and Saining Xie. A convnet for the 2020s. In *IEEE Conference on Computer Vision and Pattern Recognition*, 2022.

[67] Maxim Tkachenko, Mikhail Malyuk, Andrey Holmanyuk, and Nikolai Liubimov. Label Studio: Data labeling software, 2020-2022. URL `https://github.com/heartexlabs/label-studio`. Open source software available from https://github.com/heartexlabs/label-studio.

[68] Elad Richardson, Yuval Alaluf, Or Patashnik, Yotam Nitzan, Yaniv Azar, Stav Shapiro, and Daniel Cohen-Or. Encoding in style: a stylegan encoder for image-to-image translation. In *IEEE Conference on Computer Vision and Pattern Recognition*, 2021.

[69] Angelo Genovese, Mahdi S Hosseini, Vincenzo Piuri, Konstantinos N Plataniotis, and Fabio Scotti. Histopathological transfer learning for acute lymphoblastic leukemia detection. In *IEEE International Conference on Computational Intelligence and Virtual Environments for Measurement Systems and Applications (CIVEMSA)*, 2021.

[70] Yutong Bai, Jieru Mei, Alan L Yuille, and Cihang Xie. Are transformers more robust than cnns? *Advances in neural information processing systems*, 34:26831–26843, 2021.

[71] Mahsa Baktashmotlagh, Mehrtash Harandi, and Mathieu Salzmann. Distribution-matching embedding for visual domain adaptation. *Journal of Machine Learning Research*, 17, 2016.

[72] Yang Liu and Hongyi Guo. Peer loss functions: Learning from noisy labels without knowing noise rates. In *International Conference on Machine Learning*, 2020.

