# OpenReview forum: "WBCAtt: A White Blood Cell Dataset Annotated with Detailed Morphological Attributes"
_NeurIPS.cc/2023/Track/Datasets_and_Benchmarks — NeurIPS 2023 Datasets and Benchmarks Poster_

### Official Review · Reviewer_ZbQt · 2023-07-19
**Richly labeled and large dataset!**

**Rating:** 8
**Confidence:** 4
**Correctness:** Yes.
**Clarity:** Yes

**Strengths:**

- High quality and clinically relevant labels of white blood cell morphology
- Reasonable distribution of different attributes without any being highly skewed
- The authors explored interesting uses of this dataset (XAI, StyleGAN, etc.)

**Additional Feedback:**

Thank you for your work!

**Documentation:**

It is great.

**Ethics:**

No concerns.

**Limitations:**

The authors have already mentioned a few limitations such as only containing healthy cells and data from a single source with a single staining method. It can affect how widely this dataset will be used.

**Opportunities For Improvement:**

- The authors already detailed the annotation process in the main manuscript and supplements. However I would like to learn more about this process since these annotations are only as good as the annotators. Student being able to label 180 images/hour (aka 1 image per 20 sec) sounds very fast to me. Therefore, I would like to learn more about the number of students, the number of pathologists and etc.

**Relation To Prior Work:**

Yes.

**Summary And Contributions:**

Thank you so much for this richly labeled and large dataset! I can see this dataset being used by highly impactful and clinical relevant AI/XAI algorithms in healthcare.

---

> ### Author Response · Authors · 2023-08-16
>
>
> ### More Annotation Details
> > The authors already detailed the annotation process in the main manuscript and supplements. However I would like to learn more about this process since these annotations are only as good as the annotators. Student being able to label 180 images/hour (aka 1 image per 20 sec) sounds very fast to me. Therefore, I would like to learn more about the number of students, the number of pathologists and etc.
>
> We appreciate your interest in our annotation process. We have three experienced pathologists, each with over a decade of expertise in white blood cell recognition, actively involved in defining attributes and overseeing the annotation process. In terms of student annotators, we have a rigorous selection process. We selected 4 out of 8 applicants, based on their strong biomedical backgrounds. These students then undergo extensive training, encompassing both straightforward and complex cases. Further details about the student selection and training are discussed in response to reviewer mmqD.
>
> Regarding the annotation rate of 180 images/hour, this is achieved through the use of Label Studio, as shown in Figure 10 of the supplementary materials. The annotators need only make 11 clicks to annotate the 11 attributes for each image. This rate is reflective of the later stages of the process when annotators are well-versed with the attribute definitions and the labeling tool. For natural images, Bearman et al. [ECCV 2016] reported that "collecting image-level labels takes 1 second per class". Given our image has 11 attributes, the expected time is 11 seconds per image, which aligns with our rate. This demonstrates that our annotation speed is not unreasonably fast, considering that we have selected and well-educated the annotator to be familiar with the cells.
> - Bearman, Amy, Olga Russakovsky, Vittorio Ferrari, and Li Fei-Fei. "What’s the point: Semantic segmentation with point supervision." In ECCV 2016.
>
> ### Healthy Cells from Single Source
> > The authors have already mentioned a few limitations such as only containing healthy cells and data from a single source with a single staining method. It can affect how widely this dataset will be used.
>
> It is correct that our dataset contains healthy cells and data from a single source with a sole staining method. However, we expect the models trained on our dataset to be still applicable to unhealthy cells and other data sources with different staining methods. In Section 5.3, titled 'Interpreting Model Bias for Blood Disorders Involving WBC,' we present an example of applying the trained model to the cells of acute promyelocytic leukemia. The paragraph labeled 'Broader Applicability' (Lines 217 - 226) briefly alludes to the utilization of the trained models with other data sources, and Section A.7 in the supplementary material provides a more comprehensive discussion of this topic. In response to reviewer U8Fk, we conducted additional experiments (detailed in Section A.13 of the supplementary material) using our attributes for the detection of Acute Lymphocytic Leukemia (ALL). These experiments suggest the feasibility of employing our attributes as interpretable features for downstream clinical tasks.

---

> > ### Comment · Reviewer_ZbQt · 2023-08-29
> >
> > Thank you so much for the clarification!

---

### Official Review · Reviewer_mmqD · 2023-07-20
**WBCAtt: A White Blood Cell Dataset Annotated with Detailed Morphological Attributes**

**Rating:** 6
**Confidence:** 3
**Clarity:** Yes

**Strengths:**

+ New detailed attributes annotations of white blood cells.
+ Detailed reasoning for each attribute is provided
+ The authors discussed interesting applications of annotations.


**Additional Feedback:**

It would be useful if the authors could provide segmentation annotations as well.

**Correctness:**

There are some limitations that are mentioned in the "Opportunities For Improvement"

**Documentation:**

Yes

**Limitations:**

It would be great if authors could show some experimental results of the method which copes with the dataset annotations noises.

**Opportunities For Improvement:**

The reviewer has serious concerns regarding the annotation quality.
1) How many pathologists are involved in the study and how many years of experience do they have?
2) It looks like the pathologists are only involved in the annotation process in the case of a few ambiguous cases. Since students and research scientists are usually not well-trained and might have missed several cases, what about cases that appear non-ambiguous to the students but are actually assigned the wrong attributes?
3) Since pathologists have not reviewed all the annotations, it would be useful to get all the cells annotated by multiple annotators.
4). Unlike many related works mentioned by the authors, the dataset is not collected by the authors. Instead, they provided only the annotations whose quality has several limitations. Therefore, the reviewer believes that this paper is not up to the level of NeurIPS.



**Relation To Prior Work:**

Yes

**Summary And Contributions:**

This paper introduces interesting new annotations to existing white blood cell-related dataset by providing 11 morphological attributes. The paper provides several applications of the annotations related to interpretability and about discovering model biases.

---

> ### Author Response · Authors · 2023-08-16
>
> ### > The reviewer has serious concerns regarding the annotation quality.
> Thank you for raising the concern. Before addressing the specific comments, we would like to remind that we have implemented a rigorous and iterative process to ensure the quality. Our initial submission has already summarized this process in the "Annotation with Quality Control" section (lines 102 - 113), with additional details provided in Section A.5 "Annotation Process and Quality Control" (lines 640 - 682) in the supplementary material. These measures encompass selective annotator screening, multi-phased annotation protocols involving pathologists, and thorough validation conducted by research scientists independently of the annotators.
>
>
>
> ### Involvement of Pathologists
> > How many pathologists are involved in the study and how many years of experience do they have?
>
> > It looks like the pathologists are only involved in the annotation process in the case of a few ambiguous cases.
>
> Three pathologists, each with over 10 years of experience, are actively engaged. Their role was not limited to the review of a few ambiguous cases. They were actively engaged in discussions whenever the annotators faced any uncertainties.
>
> ### Qualification of Students and Scientists
> > Since students and research scientists are usually not well-trained and might have missed several cases,
>
> We understand the reviewer's concerns about the training of our research scientists and student annotators. Our research scientist is well-trained with a strong background in biomedical science and is highly familiar with blood cell images and relevant literature. From the onset of the project, there have been continuous and comprehensive discussions between the research scientist and the pathologists during both the attribute definition phase and the annotation process. While it's true that our student annotators may not have as much experience, we have carefully selected those majoring in biomedical science from the globally recognized School of Biological Sciences (ranked 33rd worldwide and 6th in the Asia region in Biological Sciences by QS World University Rankings in 2021 [https://www.ntu.edu.sg/sbs/about-us/our-history ]). We further narrowed our selection to four individuals, out of eight applicants, based on a rigorous screening process. These chosen students underwent extensive training sessions, followed by a trial annotation session that tested their comprehension. Feedback was given with a focus on annotation quality and adherence to standards, which prompted the students to revise their annotations, thereby ensuring accuracy. Lastly, the images we annotated were normal cell images from healthy individuals and not the irregular cases associated with blood disorders. Therefore, since the majority of the attributes we defined are frequently covered in their coursework, we believe that highly educated students can be capable of recognizing these attributes, considering the extensive training we have provided to them.
>
>
> ### Potential Errors
> > what about cases that appear non-ambiguous to the students but are actually assigned the wrong attributes?
>
> We fully acknowledge these potential errors despite our extensive quality control measures. However, any dataset contains some annotation errors. Given that no prior dataset annotated cell images with dense attributes, we believe that our dataset holds significant value. Our initial submission demonstrated three successful applications of our annotations, suggesting that any label noise is minimal. In response to the reviewer U8Fk's feedback, we have added an additional example showing how our annotations can be used in the prediction of leukemia. Furthermore, we conducted an additional experiment using a method to incorporate label noise (see below), which further suggests that the remaining annotation error, if any, is not substantial.
>
> ### Multiple Annotations
> > 3. Since pathologists have not reviewed all the annotations, it would be useful to get all the cells annotated by multiple annotators.
>
> We agree that obtaining annotations from multiple annotators is beneficial. While we were unable to carry out this process for all cells, we conducted it on 10% of the dataset. This sampling study resulted in an agreement rate of approximately 96.1%. This agreement rate is comparable to those accepted in previous works in the NeurIPS dataset track [https://openreview.net/forum?id=gud0qopqJc4, cited as 27 in our paper] (97.6%, 94%, and 92% on three small sets). Furthermore, we'd like to remind that our comprehensive validation process (line 672-676 in the supplementary material) ensures that each cell is reviewed by at least two qualified individuals.
>
> *Update*: The second annotation has been uploaded here: https://github.com/apple2373/wbcatt/tree/main/submission/re-annotation

---

> > ### Author Response · Authors · 2023-08-16
> >
> >
> > ### Not Level of NeurIPS
> > > Unlike many related works mentioned by the authors, the dataset is not collected by the authors. Instead, they provided only the annotations whose quality has several limitations. Therefore, the reviewer believes that this paper is not up to the level of NeurIPS.
> >
> > We respect diverse viewpoints, but we maintain that the fact that we did not collect the data and provided additional annotations should not diminish its NeurIPS level of quality. We are aware of two papers published in the NeurIPS dataset track that also provided additional annotations to an existing dataset. [https://openreview.net/forum?id=gud0qopqJc4, https://openreview.net/forum?id=H-d5634yVi, cited as 26 and 27 in our paper]. Naturally, the central point pertains to "the annotations whose quality has several limitations." We recognize that our annotations, like all datasets, have their limitations. We believe that our dataset still makes a significant contribution to the field, considering that no prior dataset exists annotating dense attributes. The reviewer is correct in pointing out that the related works we mentioned have collected WBC images on their own. However, none of them have provided annotations as rich and dense as ours.
> >
> > ### Method to Mitigate Label Noise
> > > It would be great if authors could show some experimental results of the method which copes with the dataset annotations noises.
> >
> > Thanks you for the suggestion. We tried the peer loss method proposed by Liu et al. (2020, ICML) and conducted two experiments.
> >
> > Experiment 1: To verify the effectiveness of the peer loss, we perturbed 10% of the annotations in the training data to artificially introduce noise. The average F1 macro scores are as follows:
> > - Without peer loss: 85.31 ± 0.24
> > - With peer loss: 88.86 ± 0.02
> >
> > The results clearly indicate that peer loss is effective in handling annotation noise in our data.
> >
> > Experiment 2: We applied the peer loss to our dataset and calculated the average F1 macro scores:
> > - Without peer loss: 91.20 ± 0.06
> > - With peer loss: 91.17 ± 0.03
> >
> > The results suggest that peer loss is not significantly effective on our data, implying that the level of annotation noise in our data is not substantial.
> >
> > We updated the supplementary material (Section A.14) to discuss the more details of these experiments.
> >
> > - Liu, Yang, and Hongyi Guo. "Peer loss functions: Learning from noisy labels without knowing noise rates." In ICML 2020.

---

### Official Review · Reviewer_sgnA · 2023-07-21
**WBCAtt: A White Blood Cell Dataset Annotated with Detailed Morphological Attributes**

**Rating:** 6
**Confidence:** 2
**Correctness:** The paper seems correct.
**Clarity:** The paper is clearly written.

**Strengths:**

1. The paper introduces a large-scale dataset of White Blood Cells (WBCs) that is densely annotated with 11 morphological attributes. These attributes provide valuable information about the cell and its components (nucleus, cytoplasm, and granules), essential for diagnosing blood-related diseases such as leukaemia and anaemia.
2. The paper provides detailed descriptions of the data collection, annotation, and analysis processes, ensuring the dataset's quality and reliability.
3. The paper demonstrates the usefulness of these attributes for various tasks, such as WBC classification, attribute prediction, and attribute-based retrieval. These tasks can help clinicians to make more accurate and informed decisions when diagnosing blood-related diseases.

**Additional Feedback:**

Please refer to the opportunities for improvement.

**Documentation:**

Yes

**Limitations:**

Please refer to above points.

**Opportunities For Improvement:**

1. The data is imbalanced in terms of attributes, e.g. (cell shape, granularity, NC ratio, e.t.c)
2. Authors should explore some baselines (ViT) other than ResNet50.
3. The authors should show results class-wise for different classes in Cell Shape, NC-ratio etc.
4.  The quality of the blood smear preparation and the staining process can influence the visibility and interpretation of certain cellular attributes. Variations in laboratory techniques can impact the appearance of WBCs under the microscope. How are the authors addressing this?

**Relation To Prior Work:**

The authors have based their work on the prior work.

**Summary And Contributions:**

This paper introduces **WBCAtt**, a novel dataset for **white blood cell (WBC) recognition** that is densely annotated with **11 morphological attributes**. These attributes are essential for explaining how hematologists recognize WBCs and diagnose blood-related diseases. The paper also conducts experiments to predict the attributes from images using a standard CNN model and showcases three applications that can benefit from the attribute annotations: **human intervention with highly interpretable models**, **counterfactual example retrieval and synthesis**, and **interpreting model bias for blood disorders involving WBCs**. The paper aims to foster advancements in **explainable artificial intelligence (XAI)** in the fields of pathology and hematology.

---

> ### Author Response · Authors · 2023-08-16
>
>
> ### Imbalanced Data
> > The data is imbalanced in terms of attributes, e.g. (cell shape, granularity, NC ratio, e.t.c)
>
> We thank the reviewer for raising this concern. We acknowledge the presence of attribute imbalance within the data, including aspects such as cell shape, granularity, and NC ratio. Nevertheless, we'd like to highlight that this attribute imbalance is inherent to WBCs. Typically, the majority of WBCs exhibit a rounded shape, except in cases where they respond to specific infections or undergo certain processes. Among the five types of WBCs, three—neutrophils, eosinophils, and basophils—contain granules and collectively account for 70% of this dataset. These cell types generally share common cytoplasm-related attributes, while being differentiated by their granule-related attributes. As a result, the ratio of cells with granules (granularity=yes) is higher than that of cells without granules (granularity=no) in this dataset. Furthermore, it's worth noting that a low NC ratio is prevalent among most cells, whereas a high NC ratio is a distinctive feature exclusive to lymphocytes.
>
> ### Other Backbones
> > Authors should explore some baselines (ViT) other than ResNet50.
>
> We appreciate the suggestion. We actually tested four backbones (VGG16, ResNet50, ViT-Base, and ConvNeXt-Tiny) during submission but didn't mention them in the main paper. Results for the three others are in Table 4, supplementary material, unintentionally unlinked from the main paper. We've now fixed this by referencing the supplementary material in Line 207.
>
> ### Class-wise Results
> > The authors should show results class-wise for different classes in Cell Shape, NC-ratio etc.
>
> Thank you for the suggestion. We have added the new Table 7 to the supplementary material, which displays precision, recall, and F-measure for each attribute value.
>
> ### Effect of Sample Preparation
> > The quality of the blood smear preparation and the staining process can influence the visibility and interpretation of certain cellular attributes. Variations in laboratory techniques can impact the appearance of WBCs under the microscope. How are the authors addressing this?
>
> We thank the reviewer for bringing this point up. We understand the potential influence of variations in staining processes and microscopic devices on the appearance of WBCs. While these variations can impact certain attributes like granule color and cytoplasm color, the majority of attributes such as cell size, cell shape, and nucleus shape remain consistent. Even for the affected ones, we note that these variations do not lead to the merging or augmentation of attribute definitions, but rather, it's a matter of mapping. To illustrate this, we created new Figure 17 in the supplementary materials. There, we provide visual explanations of how WBC appearances can differ based on various staining techniques and microscopic devices, using three datasets: PBC Dataset (used in this work), APL Dataset (referenced as [59] in our paper), and Raabin-WBC (referenced as [7] in our paper). Despite the color variations between datasets, the distinctions in color within the same dataset remain discernible. For instance, even with the variation in red granule color between Raabin-WBC and the PBC dataset, the difference between red granule color and pink granule color within the same dataset is still distinguishable.

---

> > ### Comment · Reviewer_sgnA · 2023-08-19
> > **Data Balancing**
> >
> > Thanks for responding to my concerns. Seeing the results in Table 7 suggests that some kind of data balancing strategy or loss weighting is required to make models performance fair across all the attributes. However, I have updated the ratings.

---

### Official Review · Reviewer_U8Fk · 2023-07-28
**A novel xAI dataset for white blood cell attributes.**

**Rating:** 7
**Confidence:** 3
**Clarity:** Yes, the paper is generally clearly w…

**Strengths:**

The dataset is large and annotated in detail. xAI for histology is a very relevant topic for clinical applications.

**Additional Feedback:**

None

**Correctness:**

Generally yes. Although, as I mentioned before, it would be good to see multiple backbones for benchmarking and also see how robust Grad-CAM observations are. For those it would also be good to run multiple seeds as well for each backbone and compare Grad-CAM maps, to ensure there is actual signal vs. just noise.

**Documentation:**

Yes.

**Ethics:**

No.

**Limitations:**

Yes.

**Opportunities For Improvement:**

Currently, the paper focuses on morphology through the lens of xAI. It would be interesting to see how morphological attributes can be used for downstream tasks/patient outcomes, i.e., can irregularity predict certain types of cancer? The authors themselves write they pick these attributes based on clinical relevance. Therefore it should be possible to link them to clinically relevant downstream tasks as well. This could be even only for a fraction of the annotated dataset. Or another, smaller test dataset with not only healthy patients like this study.

Single domain: The current dataset focuses on white blood cells only, while many of the described attributes could also be applied to other cell types. This would enable benchmarking more general xAI models for cells in general rather than for white blood cells only. Same applies to having datasets from multiple institutions/hospitals instead of only one.


Benchmarking is currently done only with Resnet50. It would be good to see other backbones for image encoding. It seems like Resnet50 is already quite strong. Therefore, more SOTA models might already reach an almost saturating performance on the benchmark?
Also runs should be done with multiple seeds, especially for downstream Grad-CAM analysis

**Relation To Prior Work:**

The authors re-annotated a previous dataset in more detail and make clear what the differences are.

**Summary And Contributions:**

The authors introduce a novel dataset with expert annotations of the morphology of white blood cells. The morphological attributes are annotated by experts and selected through the clinical literature. They benchmark ResNet50 on their dataset.

---

> ### Author Response · Authors · 2023-08-16
>
> ### More Clinically Relevant Downstream Tasks
> >Currently, the paper focuses on morphology through the lens of xAI. It would be interesting to see how morphological attributes can be used for downstream tasks/patient outcomes, i.e., can irregularity predict certain types of cancer? The authors themselves write they pick these attributes based on clinical relevance. Therefore it should be possible to link them to clinically relevant downstream tasks as well. This could be even only for a fraction of the annotated dataset. Or another, smaller test dataset with not only healthy patients like this study.
>
> Thank you for the suggestion. We investigated the ALL-IDB dataset (http://homes.di.unimi.it/scotti/all/, cited as [4] in our manuscript), which contains cell images of Acute Lymphoblastic Leukemia (ALL). This dataset poses a binary image classification task to distinguish healthy individuals from ALL patients. Since this dataset lacks attribute annotations, we obtained the attributes by applying the model trained on our dataset using a simple domain adaptation technique. Subsequently, we trained a gradient boosting classifier using these attributes and obtained the following accuracy results, along with the baselines reported in [Genovese et al., 2021] (the same authors as the ALL-IDB dataset):
>
> - Our attribute-based classifier: 90.77 ± 1.23
> - ViT-Base trained by us: 91.28 ± 2.17
> - ResNet18, as reported in [Genovese et al., 2021]: 88.69 ± 2.67
> - VGG16, as reported in [Genovese et al., 2021]: 87.54 ± 3.15
>
> As we can see, the attributes are indeed capable of classifying ALL cases. We updated the supplementary material (Sec. A13) to discuss this in more detail.
>
> - Genovese, Angelo, Mahdi S. Hosseini, Vincenzo Piuri, Konstantinos N. Plataniotis, and Fabio Scotti. "Histopathological transfer learning for acute lymphoblastic leukemia detection." In IEEE International Conference on Computational Intelligence and Virtual Environments for Measurement Systems and Applications 2021.
>
> ### Single domain
> > The current dataset focuses on white blood cells only, while many of the described attributes could also be applied to other cell types. This would enable benchmarking more general xAI models for cells in general rather than for white blood cells only. Same applies to having datasets from multiple institutions/hospitals instead of only one.
>
> The reviewer is correct that our dataset focused on white blood cell images from a single hospital, and many of the described attributes can be applied to other cell types. While extending the dataset to other cells or other data sources is future work, we'd like to note that models trained on our current dataset can be used for other cells or images from other sources. The paragraph "Broader Applicability" (Line 217 - 226) briefly mentions applying the trained models to other data sources, and Section A.7 in the supplementary material discusses it in more detail.

---

> ### Author Response · Authors · 2023-08-16
>
> ### Other backbones
> > Benchmarking is currently done only with Resnet50. It would be good to see other backbones for image encoding. It seems like Resnet50 is already quite strong. Therefore, more SOTA models might already reach an almost saturating performance on the benchmark?
>
> We tried four backbones (VGG16, ResNet50, ViT-Base, and ConvNeXt-Tiny) at the time of submission but only reported ResNet50 results in the main paper. We reported the results of the other backbones in the supplementary Table 4 but forgot to mention in the main paper. We added the pointer at the Line 207 in the main paper.
>
> While SOTA models have showcased impressive performance, there is still considerable room for improvement in some aspects. For example, the nucleus shape performance stands at 75.70% F1 using ResNet50, with the best-found backbone (ConvNeXt-Tiny) only slightly surpassing it at 77.97%, notably below the human agreement rate of 89.6% (Figure 11). The seemingly high average F1 score is somewhat inflated by easier binary attributes, like granularity. This arises because the overall macro F1 is the average of F1 scores across 11 attributes; hence, even a significant gain (e.g. 3%) in the challenging nucleus shape category would have limited impact on the overall metric (e.g., 0.27% from 3%) if performance of the other attributes remains the same.
>
> ### GradCAM with other backbones
> > Also runs should be done with multiple seeds, especially for downstream Grad-CAM analysis
>
> Except GradCAM, we already report numbers from three runs with different seeds, along with the mean and 95% confidence intervals.
>
> > It would be good to see multiple backbones for benchmarking and also see how robust Grad-CAM observations are. For those it would also be good to run multiple seeds as well for each backbone and compare Grad-CAM maps, to ensure there is actual signal vs. just noise.
>
> We appreciate the reviewer's suggestions. In response, we conducted additional runs with different seeds and used various backbones for benchmarking. The results, including Grad-CAM heatmaps of ResNet50 from two different seeds and other backbones (VGG16, ConvNeXt-Tiny, Vit-Base), are provided in the supplementary materials (Figure 16). Figure 16-a demonstrates consistency across all backbones, with Grad-CAM accurately highlighting the cellular region with vacuoles. In Figure 16-b, most backbones pinpoint the areas with leaked cellular substances when predicting the WBC as irregular shape. Furthermore, in Figure 16-c, most backbones fail to recognize the white blood cell (WBC), leading to predictions of a smaller cell size. This consistent pattern across most backbones aligns with the example presented in the main paper (Figure 4).

---

### Author Response · Authors · 2023-08-16

We greatly appreciate the valuable and constructive feedback provided by the four reviewers. We have finished posting the initial author responses and have uploaded the updated manuscript. The majority of the updates have been included in the supplementary material. Unlike the initial submission, this time we have placed the supplementary material directly at the bottom of the main submission PDF, eliminating the need for an additional PDF within the zip file. The new contents are indicated by blue text, which should assist the reviewers in locating them more easily.

---

### Decision · Program_Chairs · 2023-09-22

**Decision:**

Accept (Poster)

**Comment:**

This paper introduces WBCAtt, a novel dataset for white blood cell (WBC) recognition that is densely annotated with 11 morphological attributes. These attributes are essential for explaining how hematologists recognize WBCs and diagnose blood-related diseases. The group leveraged an existing dataset and added rich annotations of the WBCs to enable explanation of a classification.
The dataset highly relies on the quality of the annotation, which was well-described and appears to be robust. The development of downstream models are improved by providing explainability. Concerns from the reviewers were adequately by the authors.